# Interpreting and Analysing CLIP's Zero-Shot Image Classification via Mutual Knowledge

**Fawaz Sammani, Nikos Deligiannis**
ETRO Department, Vrije Universiteit Brussel, Pleinlaan 2, B-1050 Brussels, Belgium
imec, Kapeldreef 75, B-3001 Leuven, Belgium
`fawaz.sammani@vub.be, ndeligia@etrovub.be`

## Abstract

Contrastive Language-Image Pretraining (CLIP) performs zero-shot image classification by mapping images and textual class representation into a shared embedding space, then retrieving the class closest to the image. This work provides a new approach for interpreting CLIP models for image classification from the lens of mutual knowledge between the two modalities. Specifically, we ask: what concepts do both vision and language CLIP encoders learn in common that influence the joint embedding space, causing points to be closer or further apart? We answer this question via an approach of textual concept-based explanations, showing their effectiveness, and perform an analysis encompassing a pool of 13 CLIP models varying in architecture, size and pretraining datasets. We explore those different aspects in relation to mutual knowledge, and analyze zero-shot predictions. Our approach demonstrates an effective and human-friendly way of understanding zero-shot classification decisions with CLIP. [1]

## 1 Introduction

Contrastive Language-Image Pretraining (CLIP) [44] has catalyzed a paradigm shift in zero-shot and few-shot learning methodologies for image classification [61, 54, 31, 36, 66, 57]. CLIP consists of a vision and language encoder, both which are trained to map positive image-text pairs close together in embedding space, while pushing away negative ones. In the context of information theory, the channel which connects two information sources is referred to as the *information channel* [10], and its reliability and effectiveness is often studied through Mutual Information (MI) analysis between the two sources [53]. The training dynamics of contrastive models inherently involve a significant degree of shared knowledge between the vision and language sources, as both models must map similar points close in the embedding space. This suggests the existence of a *vision-language information channel* (Figure 1a) wherein the shared knowledge between the two modalities is stored.

Inspired by this, we aim to interpret this channel and measure the relationship and mutual knowledge between the image and text encoders of CLIP, for a given zero-shot prediction. We therefore pose the following question: *What concepts did the vision and language encoders learn in common, such that the image-text points are closer or further apart in the joint space?*

The two sources of information—the vision encoder and the text encoder—differ in modality: the vision encoder provides interpretation as visual regions, while the text encoder can only provide interpretation as text. To understand the commonalities in what both encoders learn, we must establish a shared medium for their interpretations. As a result, applying existing attribution techniques [51, 60, 45, 2] does not suffice. Moreover, the information channel is composed of *discrete* units of information (i.e., bits), however these attribution techniques provide general, high-level interpretations

---

[1]https://github.com/fawazsammani/clip-interpret-mutual-knowledge

38th Conference on Neural Information Processing Systems (NeurIPS 2024).

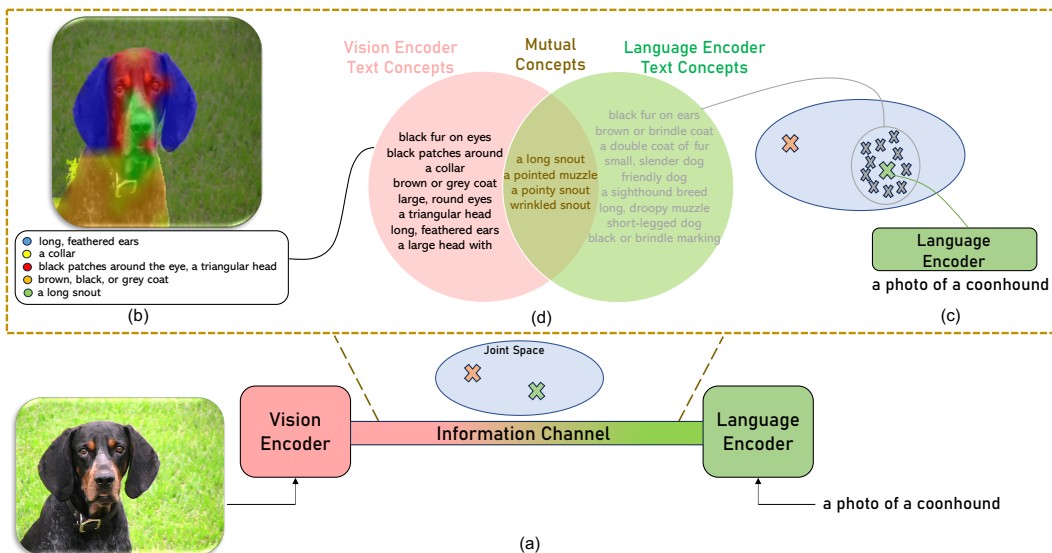

Figure 1: CLIP maps visual and textual inputs into a joint embedding space, with an information channel expressed in terms of the Mutual Information (MI) between them (**a**). We interpret the visual features from the vision encoder with multimodal concepts (**b**) which represent object parts and their corresponding textual description. From the language encoder, we identify points (shown in grey) around the zero-shot prediction (shown in green) as textual descriptions of the predicted class (**c**). By considering the textual descriptors corresponding to the visual concepts, and the textual descriptors of the language encoder for the predicted class, the two encoders establish a common space of textual concepts allowing us to identify mutual concepts and analyze their shared knowledge (**d**).

(*e.g.,* attributing the main object in the scene). They do not break-down the entangled attribution into their internal components. For instance, when applied to images of different dog breeds, attribution techniques might highlight the entire dog, indicating that the model is focusing on the correct object. However, they do not reveal what exactly in the main object influenced the model's decision. Which specific features of the dog were important? Is it the shape of the nose, the ears, the body, the head or the snout? ImageNet [30], for example, is a coarse-grained dataset, requiring models to learn distinctive concepts of an object to make decisions, but current explainability techniques do not reflect this. Therefore, we argue that distinctive fine-grained concepts are more beneficial for representing the discrete units in a channel, while also facilitating the calculation of mutual information between the two sources efficiently.

To address this question, we interpret the outcome of the visual and textual encoder as discrete random variables and use the MI to quantify the amount of information obtained about one random variable (visual data) through the other random variable (textual data). Drawing from this inspiration, we strive towards an interpretation and analysis approach of textual concepts; short descriptions in natural language (*e.g.,* "a long snout", "feathered ears"). In addition to being human-friendly interpretable, understood even to layman users, each textual concept can be mapped to an integer in a dictionary of predefined concepts (*e.g.,* "a long snout" $\rightarrow$ 0, "feathered ears" $\rightarrow$ 1). Since integers are discrete, they can represent the information units of the channel, while also facilitating the calculation of MI in the discrete space directly, which is fast, efficient, and reliable. This approach also eliminates the need for MI approximations typically required in the continuous space.

In order to achieve this, we need the two CLIP encoders to output random variables in the same space (that is, the space of textual concepts). In the vision encoder, we first refer to visual concepts as object parts grounded in the image and directly extracted from the visual features (Figure 1b, top). Those are discrete visual units that are not in the textual domain, however each of them can be described via a textual concept. Therefore, we refer to textual concepts in the vision encoder as textual descriptions of those visual concepts. As a result, multimodal concepts in the vision encoder are corresponding pairs of visual-textual semantics describing discriminative parts of an object (Figure 1b). Depending on the dataset, an object can also refer to the main scene (*e.g.,* lake or ocean in Places365 dataset [65]). Notably, our approach does not involve training any model to generate those multimodal concepts. The textual component of these multimodal concepts at the

vision encoder are now expressive of the visual concepts in the text domain. Once the mapping of visual concepts to textual concepts is achieved, we proceed with extracting textual concepts from the language encoder. This can be achieved by identifying points around the zero-shot prediction (Figure 1c). Given the output embedding of the predicted class (green point) from the language encoder, we identify related textual concepts (grey points) around that prediction. These directly serve as textual concepts explaining the prediction. The two encoders of CLIP now share a common medium of textual concepts, and we can establish the mutual concepts of both the vision and language encoders (Figure 1d). By observing Figure 1d, we see that the snout and its physical features (e.g., wrinkled, long, pointy) are expressive of what the vision and language encoders learn in common, which influence the prediction of a "bluetick coonhound" in the joint space.

Our work contributes as follows: 1) it introduces a user-friendly approach to interpreting CLIP's visual features through multimodal concepts, and we demonstrate the effectiveness of those concepts by surpassing other baselines and achieving gains of up to 3.75% in zero-shot accuracy. 2) it enables us to visualize what CLIP models learn in common when making zero-shot predictions, and how the two encoders influence each other, and 3) it allows us to explore relationships between various model aspects (model size, pretraining data, and accuracy) and its shared knowledge, and inspect the degree of correlation between the CLIP vision and text encoders.

## 2   Related Work

**Multimodal Explanations:** So far, post-hoc multimodal explanations have been limited to the context of Natural Language Explanations (NLE) [41, 24, 48]. NLEs are annotated textual explanations for the output prediction for a variety of vision and vision-language tasks, where models are explicitly trained to generate such explanations. The visual explanation counterpart is typically obtained by visualizing the attention weights of the prediction. However, there are two significant issues in NLEs. Firstly, we argue that any interpretability technique based on training is not faithful to the model being interpreted, and falls more towards the task of image captioning where the caption is the explanation. Explanations should not reflect what humans desire, but rather reflect the model's own reasoning. Training these models also involves learning biases and statistical correlations, akin to the challenges faced by any machine learning model. A recent work [47] showed that trained textual explanation models are highly susceptible to the *shortcut bias learning* problem, rendering the explanation ineffective despite achieving state-of-the-art results on Natural Language Generation metrics. Secondly, both the visual and textual explanations generated by NLEs are general, high-level and entangled (*e.g.,* highlighting the main object in the scene). On the other hand, our multimodal explanations tackle both issues outlined in NLE. They are (i) training-free and (ii) offer distinctive, fine-grained concepts. Another line of work [36, 43] extracts descriptors from a large language model and uses them as additional information when building class embedding weights of CLIP. The set of descriptors with the highest similarity with respect to the global image are considered as an explanation for the prediction. While those textual concepts are fine-grained, the explanation generated is *single-modal*. Different from [36], our concept-based explanations are multi-modal fine-grained explanations, composed of visual-textual concepts which are directly grounded in the image. Finally, [64] analyzes primitive concepts in vision-language contrastive models. We discuss this work in Section J in the appendix since it is less-relevant to our study.

**Joint Embedding Space of Contrastive Models:** A few works investigate the vision-language modality gap in the joint feature space. [18] suggests that this gap stems from inherent differences between the two data modalities. Conversely, [32] discovered that there exists a gap that causes the image and text embeddings to be placed in two distinct regions in the joint space without any overlap. In contrast to [18], they attribute this gap to the inductive bias of neural network architectures, such that embeddings of two randomly initialized models are inherently separated within the joint space, and the contrastive learning objective maintains this separation. Different from the aforementioned works, our study does not investigate the gap. Instead, we assume the gap is fundamentally present, and analyze the strength of the shared knowledge within the two models, which influence this gap.

## 3   Method

Consider the problem of image classification, where the aim is to classify an image into a set of categories $\mathcal{Y}$. For ImageNet [30], $|\mathcal{Y}| = 1,000$. CLIP [66] formulates image classification as a retrieval

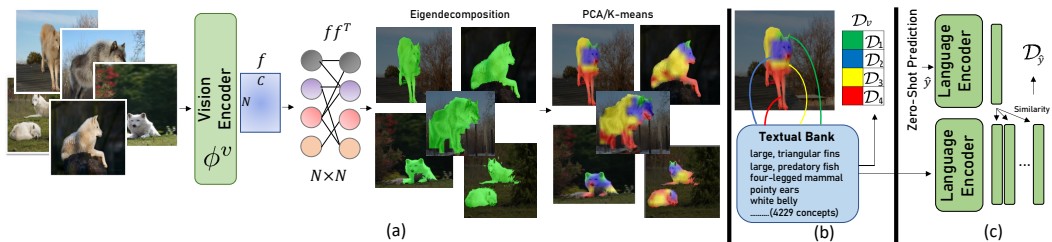

Figure 2: A high-level overview of our method for deriving visual concepts at the vision encoder (**a**), querying each visual concept individually from a textual bank to describe the visual concept in natural text (**b**), and then deriving textual concepts at the language encoder (**c**). The outputs of (b) and (c) share a common space of fine-grained textual concepts such that mutual information can be better calculated.

task by using the textual class names of $\mathcal{Y}$ denoted as $\mathcal{Y}^t$, converting them into fixed natural text prompts (*e.g.,* an image of a {class}), and encoding them with the language encoder of CLIP. The image is then encoded with the visual encoder of CLIP. The nearest category $y \in \mathcal{Y}$ to the image in the shared embedding space is then selected as the predicted class. In this context, the language encoder of CLIP can be seen as an encoder which encodes the weights of an image classifier.

**Notation:** We consider an image $I \in \mathbb{R}^{H \times W \times 3}$, a CLIP model $\phi$ composed of a Vision Transformer (ViT) encoder $\phi^v$ and a language encoder $\phi^l$, and a set of features $f = \phi^v(I) \in \mathbb{R}^{N \times C}$ extracted using $\phi^v$, where $N = H/P \times W/P$, with $P$ being the patch size of $\phi^v$ and $C$ being the feature dimension size. $\phi^v$ and $\phi^l$ are each followed by separate projection layers $\psi \in \mathbb{R}^{C \times c}$ which are fed with the [CLS] vector of the feature representation. For ease of notation, we represent the similarity score in the unified embedding space between an image-text input pair $(i, j)$ by $s(i, j) = (\psi^v \circ \phi^v(i)) \cdot (\psi^l \circ \phi^l(j))^T$. Similarly, we define $s^l(j^1, j^2)$ as the similarity in the language embedding space between two text inputs $(j^1, j^2)$ by replacing $\psi^v, \phi^v$ with $\psi^l, \phi^l$, respectively.

We utilize a Large Language Model (LLM) to generate descriptors[2] for all classes of a dataset we analyze. These descriptors are then combined into a unified set $\mathcal{D}$ which contains all class-agnostic textual descriptors (i.e., the class name does not appear in the descriptor), and its cardinality (the number of descriptors it contains) is $D$, that is, $D = |\mathcal{D}|$. For the ImageNet dataset, $D = 4{,}229$ after discarding repetitive descriptors across the entire pool. Concepts in $\mathcal{D}$ are now applicable to any object and are not restricted to the class they were extracted from. For example, the textual descriptor "can be hung from a tree" is extracted from the class "swing" in ImageNet, but can now be applied to many other classes (*e.g.,* monkey, siamang). The prompt and LLM we used, along with a detailed ablation study on various prompts and LLMs as well as the relevance and diversity of the generated descriptors, are presented in Section A.4 of the appendix.

Measuring the relationship and mutual knowledge between the image and text encoders for a given prediction is not straight-forward, as the two encoders differ in modality. Concepts in the language encoder can only be described via text, and concepts in the vision encoder can natively be described by image regions. Therefore, we need to map both concept modalities into a common space in order to quantify the mutual knowledge between the two encoders. A high-level overview of our method is shown in Figure 2. Given a set of images, we extract their visual features and perform spectral graph clustering on those features to obtain the most prominent image patches. We derive post-hoc grounded visual concepts representing object parts by applying Principal Component Analysis (PCA) or K-means clustering solely on the identified prominent patches (this is shown in Figure 2a). We encode the textual descriptors $\mathcal{D}$ with the CLIP language encoder, and query each visual concept region (encoded with the CLIP visual encoder) from the set of descriptors $\mathcal{D}$. In this way, we associate each visual concept with a textual concept describing it in natural text, producing $\mathcal{D}_v \subseteq \mathcal{D}$ (Figure 2b). In the final stage, the zero-shot predicted class and the set of descriptors $\mathcal{D}$ are encoded with the CLIP language encoder, and the most similar descriptors close to the zero-shot prediction in the language embedding space are retrieved, producing $\mathcal{D}_{\hat{y}} \subseteq \mathcal{D}$ (Figure 2c). Now, the two encoders share a common space of textual concepts, and the MI can be calculated efficiently in the discrete

---

[2]we will use the terms "descriptors" and "textual concepts" interchangeably

space using a probability-based approach by mapping each textual concept to a corresponding integer. The MI between $\mathcal{D}_v$ and $\mathcal{D}_{\hat{y}}$ is defined as:

$$I(\mathcal{D}_v; \mathcal{D}_{\hat{y}}) = H(\mathcal{D}_v) + H(\mathcal{D}_{\hat{y}}) - H(\mathcal{D}_v, \mathcal{D}_{\hat{y}}), \tag{1}$$

where $H$ represents the entropy and $H(.,.)$ represents the joint entropy which we compute through a simple contingency table. We provide the derivation for this formulation in Section B of the Appendix. In the next subsections, we describe each of the aforementioned steps shown in Figure 2. Finally, we elaborate on the formulation of mutual information dynamics in Section 3.3.

## 3.1 Multi-modal Concepts in the Visual Encoder

**Visual-based Concepts:** We first identify and separate the prominent patches in $f$ from those that are non-relevant. Drawing inspiration from prior works on unsupervised object localization [35, 55, 6], we propose to decompose the feature space $f$ into two groups, identifying the most prominent group as the focal point of the model. Specifically, we first construct an affinity matrix $A^f$ from the patchwise feature correlations of $f$: $A^f = ff^T \in \mathbb{R}^{N \times N}$, where $A^f$ serves as a spectral graph representing rich semantic information within the features. Each node in this graph corresponds to an image patch. We then apply eigendecomposition on $A^f$ and extract the second largest (non-zero) eigenvector $e_f \in \mathbb{R}^N$, known as the Fiedler eigenvector. The sign of each element in $e_f$ represents a binary segmentation mask, dividing the graph nodes into two groups with minimal connectivity. We consider the subset of patches $f^p$ corresponding to a positive sign in $e_f$ as the most prominent and obtain an importance map. This decomposition technique is simple, fast and only requires features without needing gradients. We adopt Conditional Random Fields (CRF) [28] as an interpolation technique to interpolate the patch-based importance map to the resolution of the image. This approach provides better visual results than other interpolation techniques, while also preserving the underlying importance map (see experimental proof in Section A.3 of the Appendix). Finally, we note that $f$ can be the *tokens* or *keys* of the last attention layer of the transformer. We ablate and analyze both in Section A.1 of the Appendix, and explore using PCA as an alternative decomposition technique.

Next, our aim is to derive visual concepts (*i.e.,* object parts) from the high-level prominent patches extracted in the previous step. We draw upon the methodologies from [40, 1, 9] and apply either PCA or K-means clustering solely on the identified prominent image patches $f^p$ across $B$ images. In Section 4, we report results using each of these techniques. This process dissects the prominent image patches into a set of distinct components or clusters $\mathcal{L}$ of length $L$, which express visual concepts. The visual concepts are unique, i.e., a patch can only be assigned to a single concept. An overview of this process is shown in Figure 2a.

**Describing Visual Concepts with Textual Descriptions:** We seek to link each visual concept identified in the previous step, to a textual descriptor. Initially, we encode each visual concept using the CLIP visual encoder by applying the visual prompt engineering approach proposed in [54]. This approach involves drawing a red circle around the region of interest or blurring the area outside it, in order to direct the vision encoder's attention to that specific region, ensuring it encodes that area rather than the entire image. This approach has achieved strong zero-shot performance across diverse localization tasks, greatly surpassing cropping-based approaches [59]. A subsequent work [63] verifies the effectiveness of this approach (see more details in Section F of the Appendix). We apply this technique to all the detected visual concepts in the image to yield a set of prompted images $\mathcal{I}_p = \{I_{p^1} \dots I_{p^L}\}$, where $L$ is the number of visual concepts. Next, we encode the textual descriptors $\mathcal{D}$ with the CLIP language encoder. Given that CLIP maps images and textual inputs close together in the embedding space, we find the associated top-$k$ textual descriptors for a given visual concept by simply computing the similarity between the embedding of $I_p^j$ and all textual descriptors $\mathcal{D}$: $s(I_{p^j}, \mathcal{D}_i)$, where $j$ ranges over the $L$ visual concepts, and $i$ ranges over the top-$k$ textual descriptors[3]. This results in an assignment matrix $\hat{\mathcal{C}} \in \mathbb{R}^{L \times D}$. However, we observed that, with this approach, numerous visual concepts get mapped to the same descriptor, suggesting a distribution with low entropy. To address this, we enhance the alignment of the two distributions by treating $-\hat{\mathcal{C}}$ as a cost matrix and transforming it into a permutation matrix $\Pi$ via Optimal Transport:

$$\hat{\Pi}(L, D) = \underset{\Pi \in \mathbb{R}^{L \times D}}{\operatorname{argmax}} \sum_{l \in \mathcal{L}, d \in \mathcal{D}} \Pi_{ld} \exp\left(\tau \hat{\mathcal{C}}_{ld}\right) \tag{2}$$

---

[3] we select the descriptors with scores more than 0.02 points above the 50-*th* percentile of values

where $\tau$ is a temperature parameter. We solve this optimization problem efficiently with the Sinkhorn-Knopp algorithm [56]. The top textual descriptor from each column of $\Pi$ is then selected as the descriptor for the respective visual concept represented by each row of $\Pi$. We denote the textual concepts produced by this stage as $\mathcal{D}_v$. In Section A.2 of the Appendix, we perform ablation studies on Optimal Transport and demonstrate that it achieves diversity among the different visual concepts.

## 3.2 Textual Concepts in the Language Encoder

Given the zero-shot prediction of CLIP denoted as $\hat{y}$ with $\hat{y}^t$ being a textual representation of the prediction, we can represent $\hat{y}$ as the center of a cluster in the joint space (green point in Figure 1c), with other points in that cluster (grey points) being textual concepts directly explaining the prediction $\hat{y}$. We use the same set of textual descriptors $\mathcal{D}$ (described in Section 3) to identify those concepts. We extract those textual concepts by computing the similarity between the language embeddings of the predicted class and the language embeddings of all descriptors $\mathcal{D}$, via: $s^l(\hat{y}^t, \mathcal{D})$. We select the top-$u$ descriptors with the highest similarity score as those textual concepts and denote them by $\mathcal{D}_{\hat{y}}$.

## 3.3 Mutual Information Dynamics

A simple calculation of the MI between the vision and language concepts as in Eq. (1), fails to account for the contribution of each individual information unit (i.e., concept) to the overall MI. We define that two sources have a strong shared knowledge when a source retains knowledge about the other, despite removing important information units from it. To realize this, we first organize the textual concepts of the vision encoder $\mathcal{D}_v$ in descending order based on their importance to the image, and sequentially ablate them, removing one at each step and calculating the MI (Eq. (1)) between them and $\mathcal{D}_{\hat{y}}$ after each removal step. This process generates a curve. We report the Area under the Curve (AUC) to represent the MI dynamics. The strength of the shared information can be identified by how fast the MI in a curve drops. A higher AUC indicates gradual or late drops of MI in the curve, and thus stronger shared knowledge. A lower AUC indicates sharp or early drops of MI as concepts are removed, and thus weaker shared knowledge. We note that knowledge-retaining is not attributed to redundant information units since all concepts in $\mathcal{D}$ are unique.

Finally, it is worth noting that the MI dynamics also serve as an evaluation strategy for the identified mutual concepts. By assuming that stronger shared knowledge is associated with higher zero-shot accuracy, we would expect a positive correlation between the AUC and zero-shot accuracy.

## 4 Experiments and Analysis

**Evaluation of Multimodal Concepts:** Since the multimodal concepts serve as inputs for MI analysis, we begin by evaluating these concepts to demonstrate their effectiveness and reliability. We formulate 3 baselines that adapt existing literature of single-modality concept-based explanations, to their multimodal case. MM-CBM is a formulation of Label-Free Concept Bottleneck Models [39] to the case of Multimodal Concept Bottlenecks. MM-ProtoSim is a formulation of the prototype-based ProtoSim [38] adapted to the multimodal case. We compare the performance of these baselines in Table 5 of the Appendix. The last baseline is denoted as "Feature Maps" and is a formulation of Neuron Annotation works [19, 11] to suit our case. Feature Maps identifies spatial feature activation maps as concepts. All baselines require training to generate textual concepts, and we train them on the full ImageNet training set. All baselines as well as our multimodal concepts are evaluated with 4 evaluation metrics common in the literature of XAI, namely, Insertion (higher is better) and Deletion (lower is better) [42], Accuracy Drop (low is better) and Accuracy Increase (higher is better) [7]. We provide a description of the baselines with qualitative example in Section D of the Appendix, and of the evaluation metrics in Section E of the Appendix. As seen in Table 1, our concept-based multimodal explanations outperforms all baselines except on the Insertion Metric, where the MM-CBM baseline wins. Although not within the scope of our work, Table 5 of the Appendix also shows that our MM-ProtoSim baseline achieves state-of-the-art results on concept bottleneck models, on the challenging ImageNet dataset in which many other works fail to scale to. We also show that it not only maintains standard accuracy, but significantly improves it, another phenomenon in which many previous works including LF-CBM fail to achieve. This shows the effectiveness of considering multimodal concepts for modeling discriminative tasks.

Table 1: Evaluation scores of our multimodal explanations compared to the baselines established. All use the same features, model and textual concept bank for fair comparison.

| Explanation | Requires Training | Delet.↓ | Insert.↑ | AccDrop↓ | AccInc↑ |
|---|---|---|---|---|---|
| MM-CBM | Yes | 3.147 | **3.385** | 2.634 | 1.013 |
| MM-ProtoSim | Yes | 3.149 | 3.358 | 2.665 | 0.943 |
| Feature Maps | Yes | 2.921 | 3.114 | 2.283 | 1.233 |
| Ours (PCA) | No | 2.460 | 3.168 | 1.582 | **1.849** |
| Ours (K-means) | No | **2.422** | 3.122 | **1.555** | 1.781 |

Table 2: Effectiveness and Relevancy of our multimodal concepts in boosting zero-shot accuracy of both ResNet and ViT CLIP models on the ImageNet validation set compared to baselines [36, 43].

| ResNets | Base | Ours | $\Delta$ | ViTs | Base | Ours | $\Delta$ |
|---|---|---|---|---|---|---|---|
| RN50 | 59.54 | **61.85** | +2.31 | ViT-B/16 | 67.93 | **70.28** | +2.35 |
| RN50x4 | 64.36 | **67.93** | +3.57 | ViT-B/32 | 63.28 | **65.58** | +2.30 |
| RN50x16 | 68.47 | **72.22** | +3.75 | ViT-L/14 | 74.69 | **76.74** | +2.05 |
| RN101 | 60.68 | **64.14** | +3.46 | ViT-L/14@336px | 75.49 | **77.64** | +2.15 |

Next, we show how our multimodal explanations are an effective application of CLIP prompt engineering for image classification with descriptions [36, 43], achieving gains in zero-shot accuracy of up to 3.75%. Another purpose of this experiment is to show that the multimodal concepts and descriptors identified, are a reliable source of input for mutual information analysis. We start by identifying the two most similar classes to the zero-shot prediction in the CLIP language embedding space. We then take the validation images from both of these classes, and extract multi-modal explanations using our approach. We then take the textual component of the multi-modal explanations as additional descriptors $\in \mathcal{D}$ and re-evaluate the zero-shot classification of CLIP [4]. If the detected open-set concepts are relevant to the prediction, we should expect an improvement in zero-shot classification accuracy. As shown in Table 2, this application shows significant gains in zero-shot accuracy for all CLIP models relative to the baselines [36, 43]. This demonstrates the effectiveness and relevance of the detected concepts to the CLIP model. More details about this experiment can be found in Section K of the Appendix.

**Models and Datasets:** Our MI analysis considers a wide range of CLIP models varying in architecture, size and pretraining datasets, evaluated on the full ImageNet validation split [30]. We consider the original CLIP ViT models [44]: ViT-B/16 and ViT-B/32 are base models of patch size 16 and 32, respectively; ViT-L/14 and ViT-L/14@336 are large models of patch size 14, where the later (denoted as ViT-L/14↑) is finetuned with an image size of $336 \times 336$. The aforementioned models are trained on the WIT 400M dataset [44]. We also consider additional models from OpenCLIP [22, 8] trained on DataComp [16] of 1B images and Data Filtering Network (DFN) [14] of 2B images. Both of these datasets use filtering strategies to curate clean, higher-quality data. We refer to these models with an additional suffix: -dcp and -dfn. Ultimately, we can analyze how model (and patch) size and pretraining datasets affect the information channel. We also consider the CNN-based ResNet (RN) CLIP models trained on WIT 400M: RN-50, RN-101, RN-50×4 and RN-50×16. The RN models with (×r) denote width scaling $r$. We also consider two CLIP ConvNeXt-Base models [33] from OpenCLIP, trained on LAION-400M [50] (ConvNeXt-B1), and on an aesthetic subset of LAION-5B [49] (ConvNeXt-B2). In total, our analysis comprises 13 CLIP models.

**Quantitative Analysis**: We start by examining the MI and its dynamics across models. In Table 3, we report the MI (applying Eq. 1) and AUC (as described in Section 3.3) for all CLIP models we analyze. Additionally, we include the dataset size used for training each model and its respective zero-shot classification accuracy on the ImageNet validation set [30]. We report these metrics for both PCA and K-means, which consistently show correlation. We sort the models based on their top-1 zero-shot accuracy. We remind readers that we define stronger shared knowledge based on higher AUC rather than higher MI. In Figure 5 (detailed further), we provide examples of classes that support this claim and contribute to this phenomenon. Our first observation is that AUC aligns well

---

[4]We find that some classes are overly discriminative, in which multimodal explanations of their neighboring classes introduce noise. For those classes, we simply do not introduce any additional descriptors to them.

Table 3: MI and AUC scores for different model families using PCA and K-means evaluated on the full ImageNet validation split, along with the pretraining data and Top-1 accuracy.

| Model Family | Model | Data Size | Top-1 (%) | MI | | AUC | |
|---|---|---|---|---|---|---|---|
| | | | | PCA | K-means | PCA | K-means |
| ViTs | ViT-B/32 | 400M | 61.66 | 7.40 | 7.26 | 3.61 | 3.39 |
| | ViT-B/16 | 400M | 67.70 | 7.50 | 7.44 | 3.62 | 3.53 |
| | ViT-B/32-dcp | 1B | 68.88 | 7.79 | 7.65 | 3.93 | 3.70 |
| | ViT-B/16-dcp | 1B | 73.37 | 7.68 | 7.58 | 3.99 | 3.81 |
| | ViT-L/14 | 400M | 74.77 | 7.94 | 7.89 | 4.47 | 4.37 |
| | ViT-L/14↑ | 400M | 76.23 | 7.96 | 7.93 | 4.51 | 4.44 |
| | ViT-B/16-dfn | 2B | **76.24** | **8.19** | **8.11** | **4.62** | **4.46** |
| ResNets | RN-50 | 400M | 58.42 | 7.14 | 7.20 | 3.23 | 3.32 |
| | RN-101 | 400M | 60.90 | 7.43 | 7.53 | 3.49 | 3.60 |
| | RN-50×4 | 400M | 65.28 | **7.53** | 7.58 | 3.84 | 3.90 |
| | RN-50×16 | 400M | **70.04** | 7.51 | **7.63** | **3.85** | **4.03** |
| ConvNeXTs | CNeXt-B1 | 400M | 65.36 | 6.47 | 6.66 | 2.54 | 2.80 |
| | CNeXt-B2 | 13B | **71.22** | **7.16** | **7.56** | **3.19** | **3.74** |

with accuracy, with ViT-B/16-dfn ranking top. Our second observation is that CLIP ViT models are characterized with stronger shared knowledge than CLIP CNNs (ResNets and ConvNeXts). This supports the premise that pretraining transformer models, which lack inductive biases, perform better when trained on larger datasets [12].

To further understand the effect of model size and pretraining datasets on MI dynamics, we divide the models into two families. We first fix the pretraining data and vary the model and patch size. For this analysis, we use ViT-B/16, ViT-B/32, ViT-L/14 and ViT-L/14↑ trained on WIT 400M. We show the curves in Figure 3 (left). As shown, larger models with more patches (either via a smaller patch size or a via a larger image size) correspond to higher AUC, suggesting that these models are better at encoding shared knowledge. Next, we fix the model size and vary the pretraining data. The results are shown in Figure 3 (middle). As shown, larger and higher-quality data lead to improved shared encoding on ImageNet. In Section G of the Appendix, we also perform analysis on the Places365 [65] and Food101 [5] datasets. The previous observations may be well-known and non-surprising. Nonetheless, what is noteworthy is the direct relationship established between the strength of the shared encoding (represented by the AUC) and model size, pretraining data and zero-shot accuracy. For example, we fit a linear line to the data points to approximate the AUC-accuracy relationship in Figure 3 (right), and determine the coefficients to be 11.24 and 25.97 for ViT models (blue line). This suggests that the accuracy is related to the strength by a factor of 11. Similarly, approximating the AUC-data relationship provides insights into the amount and quality of pretraining data required to achieve a desired strength of shared knowledge. This principle also extends to model size and could allow us to design small, efficient models. Finally, this relationship is valuable for model selection, especially in cases where accuracy alone is insufficient for distinguishing between models (e.g., models with very similar accuracies such as ViT-L/14↑ and ViT-B/16-dfn). In Table 3 and Figure 3 (green line), we show that AUC demonstrates a linear correlation with accuracy and model size within the ResNet family. It is worth noting that AUC-Accuracy relationship only holds *within* a given architecture; different architectures (ViT, ResNets, ConvNeXts) have different designs and ways of learning representation. Therefore, they differ in how they utilize data and encode shared knowledge. As an example, RN-50×16 includes much more dimensions to store information than ConvNeXt-B2, and it is reasonable to expect that the shared information in RN-50×16 is stronger, despite ConvNeXt-B2 achieves a slightly higher accuracy. In fact, the gradual design trajectory of ConvNeXt [33] was built and tuned towards a higher classification accuracy.

**Analyzing Concepts in the Vision Encoder:** Although the focus of our work is to interpret and analyze mutual concepts in the vision and language encoders of CLIP, we can still utilize our multimodal explanations to inspect internal concepts learned in the vision encoder of CLIP for individual instances. Figure 4 shows 4 examples, each represented by a distinct visual cluster denoted by a different color. The corresponding textual description for each visual cluster is provided below, aligned with its corresponding color. Different from attribution-based techniques [51, 60, 45, 2] which typically highlight high-level and general features, our multimodal explanations disentangle the features to offer visually and textually distinctive, fine-grained concepts. An example of such

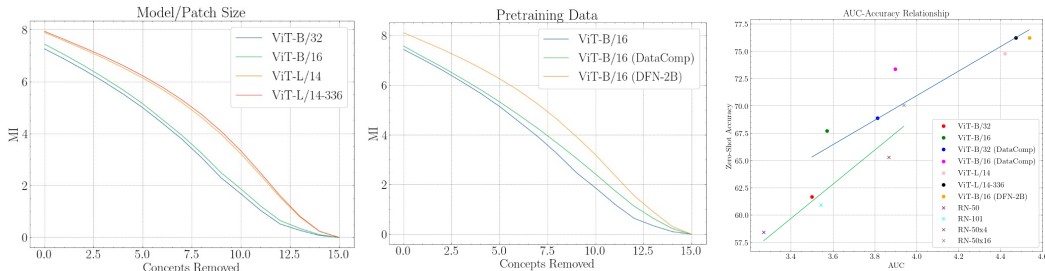

Figure 3: MI Dynamics curve comparing model families (**left**) and pretraining datasets (**middle**). Correlation of AUC with zero-shot classification accuracy is shown **right** for ViTs and ResNets.

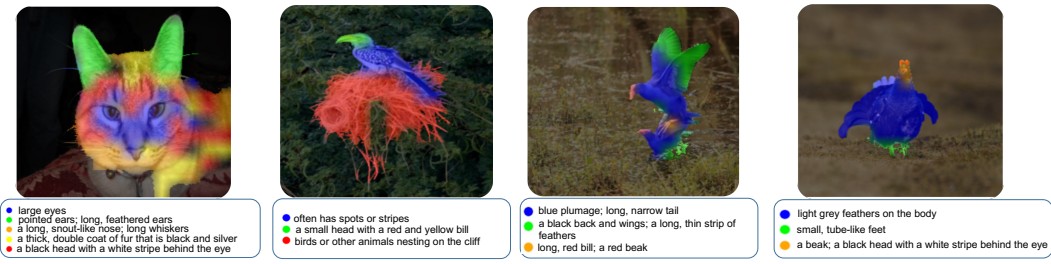

Figure 4: Qualitative examples of multimodal concepts in the vision encoder. The second-top textual descriptor may be omitted to avoid clutter.

concepts in Figure 4 are long feathered ears, long whiskers (first example); blue plumage, red beak (third example). These types of concepts are significantly more beneficial for understanding models and analyzing MI. More examples of our multimodal concepts are in Section H of the Appendix.

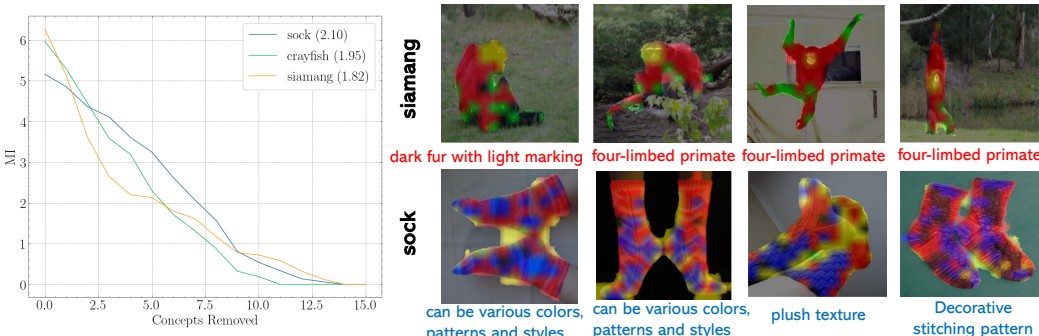

Figure 5: Analyzing concepts in different ImageNet classes with Mutual Knowledge

**Analyzing Mutual Knowledge across Classes:** Next, we show how to make use of our multimodal explanations and MI dynamics to analyze concepts across a set of images pertaining to a class. In Figure 5 we show examples of three classes from ImageNet: *sock, siamang* and *crayfish*. Results are averaged across all $B$ images of the class ($B = 50$ for ImageNet validation set). On the left, we show the MI curves. Note that, although the initial MI value for *siamang* is higher than that of *sock*, the AUC for *sock* (2.10) surpasses that of *siamang* (1.82). This is attributed to the fact that the curve for *siamang* starts at a higher point but drops faster at early stages. This shows that considering MI alone without its dynamics, is not representative of the strength of shared information. Next, we aim to delve deeper into this analysis using our multimodal explanations. To accomplish this, we examine the same semantic concept corresponding across all images. In Figure 5 (right), we showcase the semantic concept representing a *gibbon body* for the "siamang" class (e.g., dark fur with light markings; four-limbed primate), visually identified by the red color. Similarly for the "sock" class, we show the semantic concept of *patterns and styles* (e.g., various colors, patterns, and styles; plush texture; decorative pattern), visually identified by the blue color. Analyzing the curves and the area under them suggests that general concepts such as patterns and styles are better encoded

than discriminative concepts such as the body of a siamang. This rationale stems from the fact that CLIP was trained on a dataset from the internet, which is less discriminatory; it is less common to encounter images of an endangered specie of a gibbon compared to general concepts such as patterns and styles, which are prevalent characteristics across many objects in the world.

**Visualizing Mutual Concepts:** Finally, we provide visualizations of the mutual concepts detected by both vision and language encoders of CLIP in Figure 6. In the first example, we see that mutual concepts are distinctive to the zero-shot prediction of celo (*e.g.,* handheld musical instrument, strings stretched across the head, a sound hole), suggesting that both encoders effectively represent the image and class in the joint space. In the second example, we see that the language encoder is stronger than the visual encoder at encoding the concept of a rattle snack since it provides related concepts, while the mutual concepts are weaker (only one mutual concept describes a rattle snack). These visualizations help us understand the common concepts learned by both encoders and how the encoders influence each other in the joint space. More examples are in Section I of the Appendix.

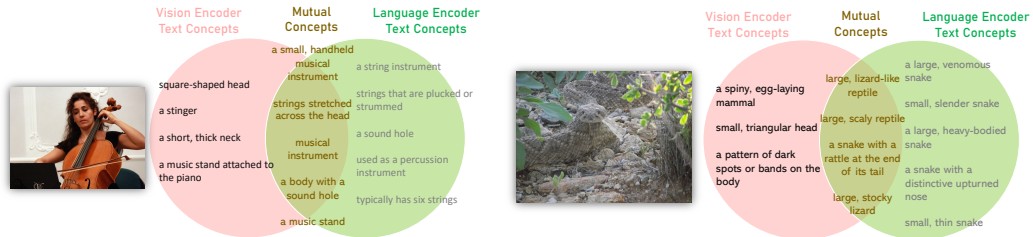

Figure 6: Visualizing Vision-Language-Mutual Concepts

## 5   Conclusion

We proposed an approach for interpreting CLIP models for image classification from the perspective of mutual knowledge, by analyzing and interpreting the information channel established along with its dynamics. In the future, our work could be extended to non-contrastive models, or even to two parts of the same model. Finally, it is important to note that, like any research, our work has its own set of limitations, which are discussed in Section C of the appendix.

## Acknowledgement

Fawaz Sammani is fully and solely funded by the Fonds Wetenschappelijk Onderzoek (FWO) (PhD fellowship strategic basic research 1SH7W24N). N. Deligiannis acknowledges support from the Francqui Foundation (2024-2027 Francqui Research Professorship on Trustworthy AI) and the "Onderzoeksprogramma Artificiele Intelligentie (AI) Vlaanderen" programme.

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

# Appendix

## A  Ablation Studies

### A.1  Feature Facets

We consider three types of vision features $f$: the *tokens* and *keys* of the last attention layer of the transformer, as well as an *ensemble* of them. In Figure 1, we show the similarity of the [CLS] token across all layers for both the tokens and key features of a ViT-B/16. As shown, the token features of the last layer are dissimilar to all previous layers, while the similarity of the key features is consistent across all layers. This is rational since token features (a function of the key features) have to adapt their feature space to align with language features. However, by running object localization experiments on the full ImageNet validation split, we find that this phenomenon is only present in large models. In Table 1 we report the CorLoc metric where we fit a bounding box around the most prominent region and calculate the percentage of samples with an IoU larger than 0.5 between that box and the ground-truth bounding box. In general, key features are more stable. By further examining the results, we note that PCA performs significantly worse than Graph Decomposition (GDC) of the feature affinity matrix, suggesting the presence of complex features that cannot be captured through linear combinations of dimensions, requiring graph-based approaches to model higher-order relationships. Therefore, we adopt eigenvector decomposition of the graph affinity matrix as the primary method for extracting prominent patches.

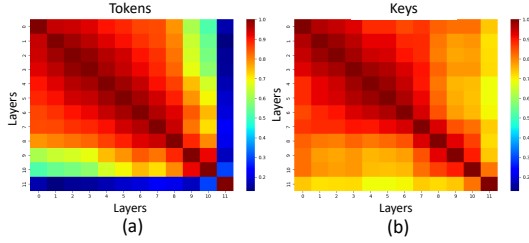

Figure 1: CLIP layer similarity analysis for the token features **(a)** and key features **(b)**.

Table 1: Ablation studies on **CorLoc** for different feature facets and decomposition methods. GDC: Graph Decomposition.

|  | ViT-B/16 | ViT-L/14↑ |
|---|---|---|
| Keys (GDC) | 54.0 | **46.9** |
| Keys (PCA) | 46.9 | 39.3 |
| Tokens (GDC) | **55.8** | 30.7 |
| Tokens (PCA) | 1.0 | 30.0 |
| Ensemble (GDC) | 54.7 | - |
| Ensemble (PCA) | 15.2 | - |

### A.2  Optimal Transport

We test the effectiveness of OT on our multimodal explanations. Let $T^l$ represent the detected textual concepts in an image $I$ for a visual concept $l$, where $1 \leq l \leq L$ such that $T = \{T^1, T^2, \ldots, T^L\}$. Since visual concepts describe unique semantic regions (*i.e.,* parts of objects), the textual descriptors associated with each visual concept should also be unique. To evaluate this, we define the Entropy metric which measures the diversity of $T$ across all $L$ visual concepts. This metric penalizes textual concepts that appear repeatedly across two or more visual concepts. A higher value indicates greater diversity. We present the findings in Table 2. The baseline case maps textual descriptors corresponding to the visual concepts without any post-processing steps. As shown, this leads to a low entropy and non-diverse results where some visual concepts are mapped to the same textual descriptor. Figure 2 shows this effect qualitatively. Using OT alleviates this issue and considerably increases the entropy.

Table 2: Entropy scores for ablating Optimal Transport.

|  | PCA | K-means |
|---|---|---|
| w/o Optimal Transport | 1.70 | 1.70 |
| w/ Optimal Transport | **2.33** | **2.34** |

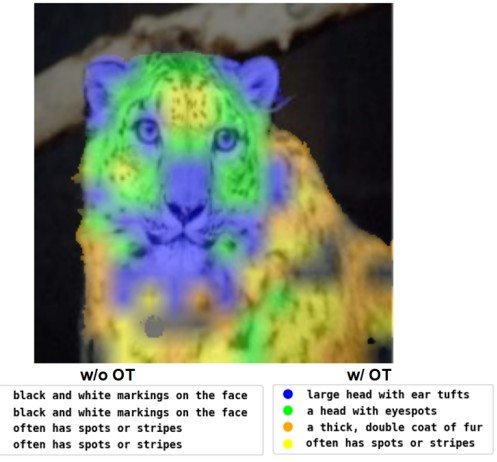

Figure 2: Optimal Transport (OT) diversifies textual concepts across their visual counterparts

## A.3    Eigendecomposition and CRF effect

To match the dimensions of the importance map with those of the image, we need to perform interpolation. However, interpolation usually results in pixel overshooting effects. As an alternative interpolation technique, we use Conditional Random Fields (CRF) [28] to interpolate the patch-based importance map to a finer-grained pixel-based map. Prior to this, we remove noisy scattered patches by extracting the largest fully-connected component (FCC) of the importance map, which represents the largest primary connected region. A recent work [15] shows that this approach enhances explainability. It is important to note that both the FCC and CRF preserve the original importance map's integrity and do not compromise its faithfulness. This is quantitatively demonstrated in Table 3 for several ViT and ResNet CLIP models. We first evaluate the effect of interpolating the binary importance map produced by eigendecomposition using nearest neighbor interpolation. To perform this evaluation, we blur the non-prominent regions of the image (those with a negative sign in $e_f$) using an $11 \times 11$ Gaussian kernel, effectively removing the details from these areas. We opt to blur those areas rather than zeroing them out in order to avoid the Out-Of-Distribution (OOD) problem reported in [21, 46]. We then reassess the zero-shot accuracy. As shown, using only the most prominenet features identified by eigendecomposition does not drastically reduce the zero-shot accuracy, with a loss of only 3% on average. This validates that the features obtained by eigendecomposition are indeed the most important and are the ones that mainly (largely) contribute to the model's performance. Next, we show the effect on zero-shot accuracy when taking the FCC, followed by CRF. As shown, this results in very marginal effects on accuracy compared to Nearest Neighbor interpolation, indicating that these components does not alter the underlying importance map. Note that the marginal effect can be either positive, where the FCC enhances zero-shot accuracy, or negative, where excluding regions other than the fully-connected component reduces accuracy. In both scenarios, the effect is very minimal and can be considered negligible. Therefore, we decided to use these two components to achieve better visualizations.

Table 3: First two columns: Baseline ImageNet validation zero-shot accuracy **(Base)** and the new accuracy after removing the non-prominent regions from the Nearest-Neighbor Interpolated importance map **(NN Interp.)**. Last three columns: Taking the fully-connected component **(FCC)** of the importance map followed by CRF interpolation leads to very marginal effects compared to **(NN Interp.)**, indicating that these components do not alter the underlying importance map

| Model | Base | NN Interp. | +FCC | +CRF | $\Delta$ |
|---|---|---|---|---|---|
| **ViT-B/32** | 61.66 | 58.09 | 58.13 | 58.22 | +0.13 |
| **ViT-B/16** | 67.70 | 64.40 | 64.30 | 64.32 | - 0.08 |
| **ViT-L/14** | 74.77 | 72.80 | 72.75 | 72.79 | - 0.01 |
| **RN50** | 58.42 | 54.41 | 54.40 | 54.43 | +0.02 |
| **RN101** | 60.90 | 56.83 | 56.84 | 56.87 | +0.03 |

## A.4 Prompt and LLM Analysis

Since the textual descriptors $\mathcal{D}$ are considered as the discrete units that mimic the vision-language information channel, it is essential to test their quality. In our work, we use the descriptors provided by [36]. This work provides a set of descriptors for each different dataset. To generate these descriptors, the work uses GPT-3.5 with the following prompt: *What are useful visual features for distinguishing a category name in a photo?*. This work also uses an in-context example to instruct the LLM to generate structured descriptors (short, distinctive). We generally find that the generated descriptors are of good quality. To show this, we have conducted an ablation study on different prompts, as well as different LLMs, using the ImageNet dataset. For each (LLM, prompt) experiment, we measured the following:

- **Zero-shot top-1 and top-5 accuracy:** These measure the relevancy of the descriptors to CLIP, and a higher accuracy implies more relevant descriptors to the class.
- **Inter-Class Diversity (InterDiv):** Measures the diversity of descriptors across different classes rather than across a single class.
- **Intra-Class Diversity (IntraDiv):** This is the cosine similarity between the different descriptors of a given class, averaged over all ImageNet classes. We used the Sentence Transformer language encoder to encode the descriptors. Note that, the lower the similarity is, the more diverse the descriptors are. Therefore, lower is better.

We considered 4 LLMs: GPT-3.5, GPT-4o-mini, GPT-4o, and the latest Llama3.1-8B-Instruct. We also considered an ensemble of 2 LLMs: GPT-3.5 and GPT-4o-mini, where GPT-3.5 provides context to GPT-4o-mini, and GPT-4o-mini answers according to its own knowledge as well as the context. Moreover, we considered 4 prompts (P):

- **P1:** "What are useful visual features for distinguishing a category name in a photo?"
- **P2:** "What are the distinctive and physical features of a category name?"
- **P3:** "What specific attributes distinguish a category name?"
- **P4:** "Which physical features and attributes make a category name different from others of the same type?"

The results for a ViT-B/16 are shown in Table 4:

Table 4: Results of different LLMs and prompts using ViT-B/16.

| Prompt | LLM | Top-1 | Top-5 | InterDiv | IntraDiv |
|--------|-----|-------|-------|----------|----------|
| P1 | GPT-3.5 | 67.93 | 91.45 | 0.345 | 0.206 |
| P1 | GPT-4o-mini | 68.39 | 91.74 | 0.236 | 0.172 |
| P1 | GPT-4o | 68.42 | 91.66 | 0.246 | 0.175 |
| P1 | Llama3.1-8B-Instruct | 68.19 | 91.56 | 0.263 | 0.184 |
| P2 | GPT-4o-mini | 68.35 | 91.69 | 0.236 | 0.164 |
| P3 | GPT-4o-mini | 68.39 | 91.78 | 0.231 | 0.152 |
| P4 | GPT-4o-mini | **68.56** | **91.83** | **0.228** | **0.151** |
| P4 | GPT-3.5 + GPT-4o-mini | 68.40 | 91.68 | 0.236 | 0.159 |

We found that P4 with GPT-4o-mini provides the best results in terms of all metrics. However, the effect is very marginal (e.g., 0.63 accuracy improvement, and 0.11 diversity improvements). Therefore, the experiment we used in our work (P1, GPT-3.5) is reliable.

## B Derivation of Mutual Information

For ease of notation, denote the set of textual concepts at the vision encoder $D_v$ by $X$, where $|X| = L_v$, and the set of textual concepts at the language encoder $D_g$ by $Y$, where $|Y| = L_T$. We remind readers that those concepts are mapped to discrete indices. The MI between X and Y is therefore:

$$I(X;Y) = H(X) + H(Y) - H(X,Y) \tag{3}$$

The entropy is defined as:

$$H(X) = -\sum_{i=1}^{L_v} p(X_i) \log(p(X_i)) \tag{4}$$

In our case, since each unit in $X$ is unique, we have a uniform distribution with equal probability: $p(X_i) = \frac{1}{L_v}$, and Eq. (3) becomes:

$$
\begin{aligned}
H(X) &= -\sum_{L_v} \frac{1}{L_v} \log\left(\frac{1}{L_v}\right) \\
&= -\sum_{L_v} \frac{1}{L_v} [\log(1) - \log(L_v)] \\
&= -\frac{1}{L_v} \sum_{L_v} -\log(L_v) \\
&= \frac{1}{L_v} \sum_{L_v} \log(L_v) \\
&= \frac{1}{L_v} L_v \log(L_v) \\
&= \log(L_v)
\end{aligned}
\tag{5}
$$

Similarly, each unit in $Y$ is unique, and we have a uniform distribution with equal probability $p(Y_i) = \frac{1}{L_T}$. In a similar manner, we obtain:

$$H(Y) = \log(L_T) \tag{6}$$

$H(X,Y)$ is obtained through a contingency table $T$:

$$
T_{ij} = \begin{cases} 1 & \text{if } X_i = Y_j, \quad \forall i \in L_v, \forall j \in L_T \\ 0 & \text{otherwise} \end{cases}
$$

$$p(X_i, Y_j) = \frac{T_{ij}}{\sum_{i,j} T_{i,j}} \tag{7}$$

$$H(X,Y) = -\sum_{i=1}^{L_v} \sum_{j=1}^{L_T} p(X_i, Y_j) \log(p(X_i, Y_j)) \tag{8}$$

Therefore, Eq. (3) reduces to:

$$I(X;Y) = \log(L_v) + \log(L_T) + H(X;Y) \tag{9}$$

## C   Limitations

Every research work is accompanied by its own set of constraints and limitations that should be acknowledged and considered. We highlight two limitations of our work. The first is in the multimodal explanations of the visual encoder. We find that in some cases, the visual concepts identified by PCA/K-means are noisy and scattered around different parts of the image. We show 4 examples of this in Figure 3. This in turns leads to the divergence of the textual concept associated to the noisy visual concept. We approach this problem by considering the largest connected region of that visual concept as input to the corresponding textual concept identification process. While this

alleviates the problem, it still presents visually unappealing results to the user. We tried to address this problem by considering better clustering techniques such as DBSCAN [13] and Hierarchical Clustering [37], but this did not show any improvements. We also noticed that this problem is less severe in self-supervised vision models such as DINO [6, 40].

Another limitation of our work is that it only analyzes zero-shot image classification tasks. However, there are several tasks that can also be performed with CLIP such as image-text retrieval, image segmentation and object localization. We leave these tasks to future work.

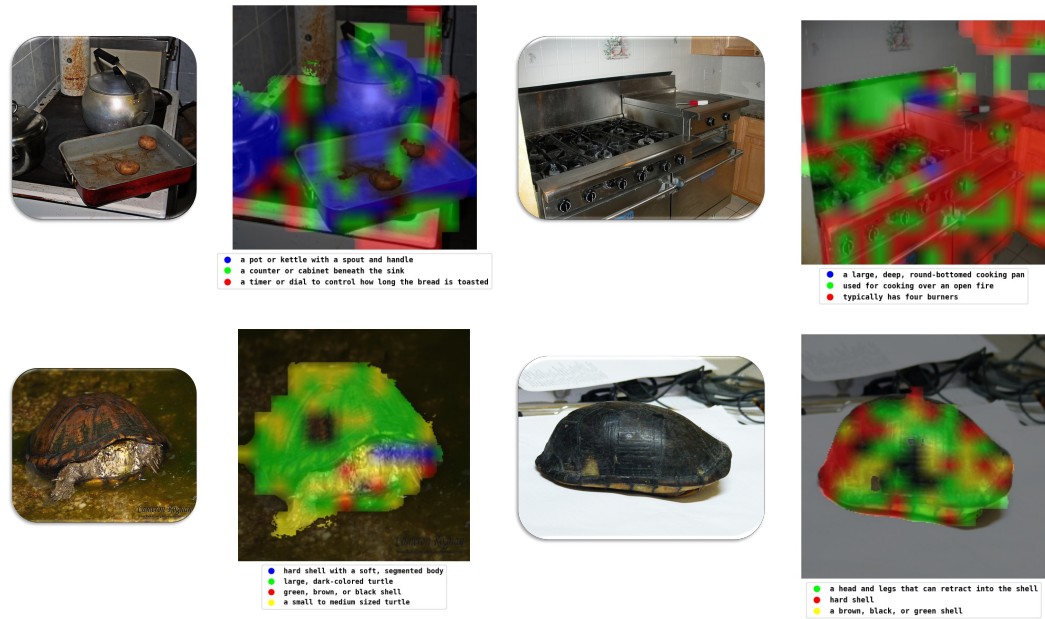

Figure 3: Noisy Scattered Visual Concepts

# D    Baselines for Concept-based Multimodal Explanations

In this section, we discuss the baseline methods we formulate for comparing against our multimodal explanations in the vision encoder. We describe and analyze these baseline here.

### D.1    Multimodal Concept Bottlenecks

Concept Bottleneck Models (CBMs) [27, 39] are networks which aim at training an inherently interpretable network, and typically sacrifice performance (e.g., accuracy) for interpretability. Given a set of features, they first predict an intermediate set of predefined concepts $\mathcal{D}$, and then use $\mathcal{D}$ to predict the final output (*e.g.,* classification) through a linear layer. Since a linear layer is inherently explainable, one can explain the prediction by simply identifying concepts in $\mathcal{D}$ with high weights to the prediction. Note that in the case of a linear layer, the weights are also the gradients with respect to the linear layer's input.

We first note that this line of work differs from our multimodal explanations; CBMs train a model to be inherently interpretable, while we explain an already pretrained model without any training. Some later works involve solely training a linear classifier on top frozen features of a given model, especially in the realm of CLIP models. One of such works is Label-Free Concept Bottleneck Models (LF-CBM) [39]. This work learns concept bottlenecks for the CLIP model: First, the CLIP similarity between an image $I$ and each concept in $\mathcal{D}$ is computed to yield $U \in \mathbb{R}^D$ where $D$ is the number of concepts. Next, a linear classifier is trained on top of $U$ to classify images. However, this does not make CBM explanations faithful to the existing frozen model which we extract features from (*e.g.,* CLIP), since training a linear classifier involves modifying the decision-making rule that the existing model was trained on (zero-shot image classification via retrieval in case of CLIP). Training a new classifier to predict targets from a set of concepts allows us to understand what concepts the

new classifier learned, and not what the existing model learned. In fact, the work of [64] (further discussed in Section J) shows that linear classifiers utilize completely irrelevant textual concepts to make predictions, highlighting that re-training a linear classifier results in a complete transformation in the decision-making process of the original model to be explained. Finally, CBMs are typically uni-modal involving one type of modality (*e.g.,* textual concepts).

Nevertheless, for comparison purposes, we assume a different scenario where we train an interpretable CBM on top of CLIP features, and modify LF-CBM to suit multimodal concept bottlenecks. We formulate two baselines which follow exactly the same architecture, shown in Figure 4, but differ in how the attention mechanism is defined. We discuss its two variants below.

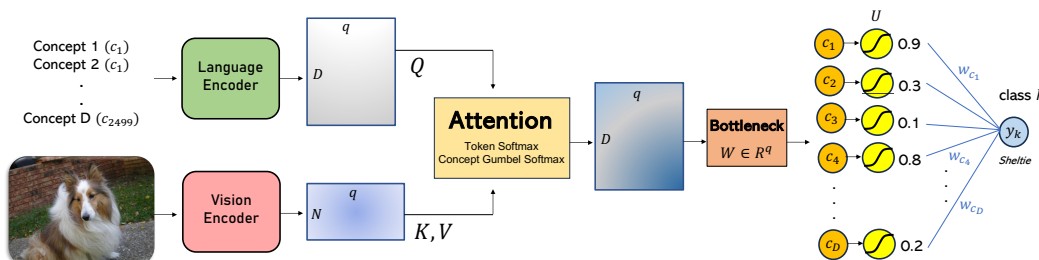

Figure 4: The MM-CBM baseline we formulate

**Multimodal Concept Bottleneck Models (MM-CBM):** We encode the set of textual descriptors $\mathcal{D}$ using the CLIP language encoder $\phi^l(\mathcal{D})$ followed by a linear projection to yield a set of concept features $Q \in \mathbb{R}^{D \times q}$ where $D$ is the number of concepts. $Q$ acts as the query to the attention mechanism. We further encode the image $I$ using the CLIP vision encoder $\phi^v(I)$ followed by a linear projection to yield a set of features $K = V \in \mathbb{R}^{N \times q}$ which act as the keys and values to the attention mechanism. Note that we use the same facet of features and concept descriptors as in our multimodal explanations. We then apply cross-attention over the tokens. Specifically, given the attention scores $QK^T \in \mathbb{R}^{D \times N}$, we apply a softmax operation over $N$ and extract a weighted summation of tokens for each concept: $\mathrm{softmax}(QK^T)V$. The attention output is then fed into a linear layer $W \in \mathbb{R}^q$ which outputs a single value for each concept in $\mathcal{D}$. We term this as the bottleneck output $U \in \mathbb{R}^D$. In LF-CBM, as mentioned earlier, the bottleneck output $U$ is determined using the cosine similarity where the values fall within the range of 0 to 1. To achieve a similar affect, we employ the sigmoid activation function on $U$. Finally, $U$ is fed to a classifier layer over all ImageNet classes. We extract the textual concepts by decomposing the prediction into its elements before the summation: $U^p = U \odot w_p$ where $w_p \in \mathbb{R}^D$ are the weights of the predicted class $p$. We then take the textual concepts corresponding to the highest values of $U^p$. For each textual concept, we take the attention weights over its visual tokens $N$ as the corresponding visual concept.

**MM-ProtoSim:** We recall that our visual concepts are unique; a pixel can be assigned to only one concept. Moreover, the textual concepts are representative of their visual concepts, but exist in a different modality. To incorporate this uniqueness into MM-CBM, we draw inspiration from ProtoSim [38] and enforce a hard assignment of a visual token to one of the $D$ concepts. Given the attention scores $QK^T \in \mathbb{R}^{D \times N}$, we apply a hard attention over the $D$ textual concepts. In practice, we relax the discrete distribution and use Gumbel-Softmax [23]. This method involves adding values sampled from the Gumbel distribution, which models maximums, to the attention logits before softmax. Therefore, we have $\mathrm{gumbel\text{-}softmax}(QK^T)V$. We extract textual and corresponding visual concepts using the same approach as MM-CBM.

We train these baselines on the full ImageNet training set, and report the Top-1 and Top-5 accuracy results on the ImageNet validation set in Table 5. The standard non-CBM baseline involves training a MLP to classify images using solely vision features. Although not within the scope of our work, we find that our multimodal baselines greatly outperforms the recent state-of-the-art LF-CBM [39] on the challenging ImageNet dataset, a dataset which many other CBM works fail to scale to. We also show that our MM-CBM not only maintains standard accuracy, but significantly improves it, another phenomenon in which many previous works including LF-CBM fail to achieve. This shows the effectiveness of considering multimodal bottlenecks for modeling discriminative tasks. The baselines

are trained using the Adam optimizer [25] with a batch size of 64 and a learning rate of 1e-4 decayed using a cosine schedule [34] to 1e-5. We set $q = 512$.

Table 5: ImageNet validation accuracy on our formulated multimodal baseline models compared to LF-CBM and Standard baseline. All models use the same CLIP ViT-B/16 model. Vis: Vision Features, Lan. Language Features, MMB: Multimodal Bottlenecks

| Model | Vis. | Lan. | MMB | Top-1 | Top-5 |
|---|---|---|---|---|---|
| Standard | ✓ | ✗ | ✗ | 73.36 | 92.75 |
| LF-CBM [39] | ✓ | ✓ | ✗ | 71.95 | - |
| MM-CBM | ✓ | ✓ | ✓ | 77.88 | 94.96 |
| MM-ProtoSim | ✓ | ✓ | ✓ | **78.79** | **95.43** |

## D.2   Neuron Annotation

Another line of work [3, 19, 11] investigates individual neuron labeling. These works annotate the functions of a subset of neurons across various layers of a given neural network with human-friendly textual concepts, and then perform quantitative analysis to examine the types of concepts that the model globally learns. This line of work falls into the category of global model explanations. MILAN [19] involves feeding in a huge set of images (in the order of millions) to a model and inspecting which set of images does each neuron, in a defined subset of neurons, respond to. Subsequently, an intensive manual human-labor process is performed to annotate responding neurons with textual descriptions. Finally, an autoregressive model is trained to generate these descriptions based on the feature activations of a given neuron across multiple layers. Ultimately, this process defines the function of a neuron by assigning a textual concept to it.

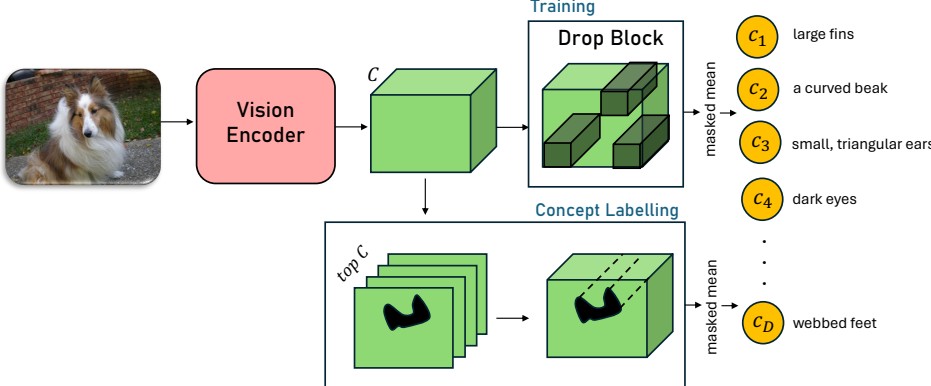

Figure 5: Training and Concept Labeling processes of the concept labeler module used in our baseline to label a feature activation map with a textual description

Similar to all issues discussed in Section 2 in the main paper concerning Natural Language Explanations, neuron annotation methods require training. This renders these methods unfaithful to the model and more akin to the task of image captioning, with the exception that they are trained on a subset of features from different layers. This is especially evident in DeViL [11] where an autoregressive generator is trained on the CC3M image captioning dataset [52] using a subset of features from different layers implemented by applying Dropout. Furthermore, these models capture biases and statistical correlations between the features and descriptions, despite achieving high evaluation scores on Natural Language Generation metrics. This phenomenon is evident by three key observations from the literature. **Firstly**, [47] highlighted that trained textual explanation models are characterized with the shortcut bias learning problem, rendering the textual explanation ineffective. **Secondly**, there exists a disparity in performance when a neuron text generator trained on MILAN annotations [19] using features from one network (*e.g.*, ResNet-101), is used to explain features of a different network (*e.g.*, ViT). In such scenarios, the achieved results are notably low, often approaching levels of 30%

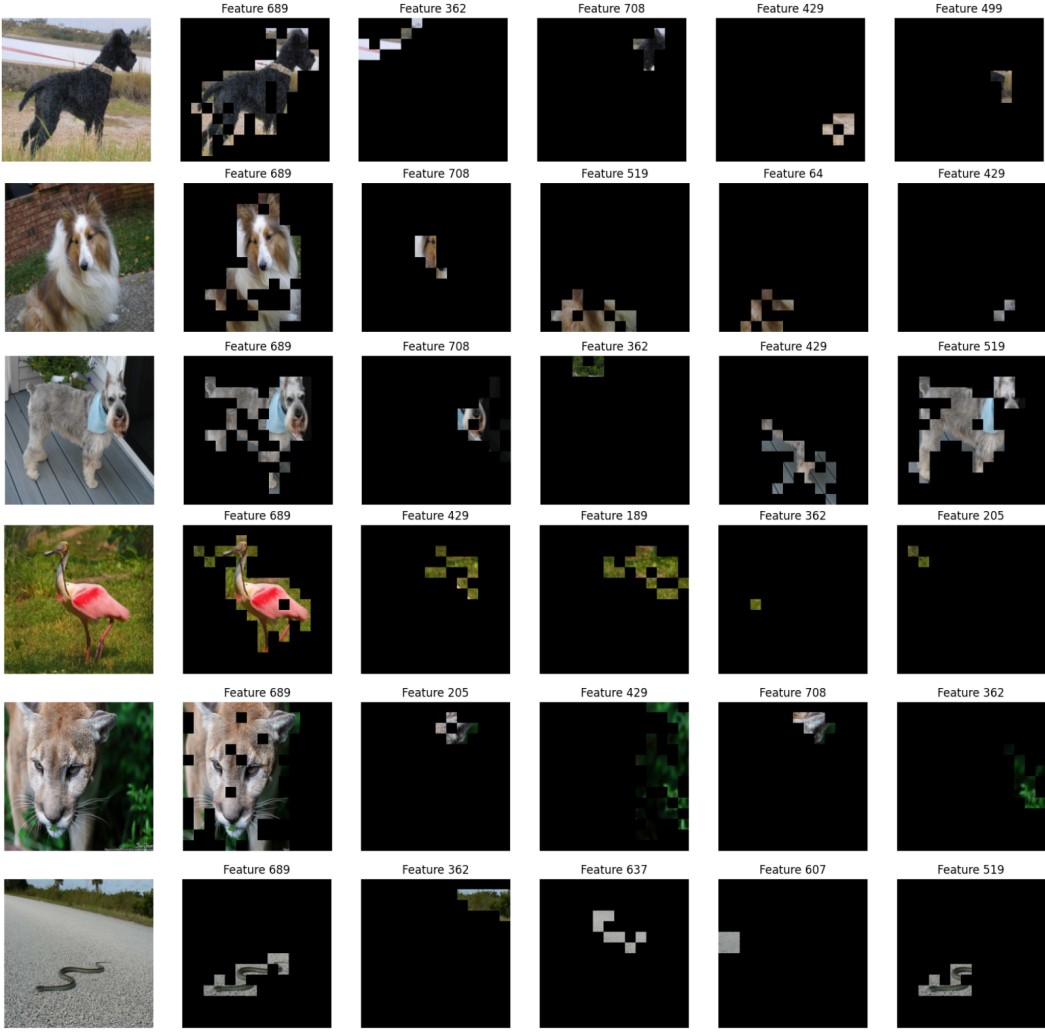

Figure 6: Feature activation maps of different neurons of the ViT-B/16. The left-most activation map represents the highest spatial norm. Neuron 689 with the highest norm always encodes the main object, while other neurons encode different concepts

accuracy, owing to discrepancies in feature spaces. This discrepancy underscores the fundamental premise that feature distributions cannot be assumed similar across models. **Thirdly**, the work of [64] shows that even non-autoregressive models such as simple linear classifiers use completely irrelevant textual concepts at the bottleneck to make predictions, showing that training renders the extracted textual concepts as ineffective. Another challenge in Neuron Annotation arises from the impracticality of annotating all neurons within a model. Due to this constraint, only a fraction of neurons can feasibly be annotated. For example, a ViT with 12 layers contains 2048 neurons in each of its hidden MLP layers, amounting to 24,576 neurons in total. However, MILAN [19] annotates only 1200 neurons (4% of the total MLP neurons). Consequently, these neuron annotation techniques offer a narrow view of the model's global behaviour.

Although this line of work differs from our multimodal explanations, for comparison purposes, we modify these works to suit our scenario and formulate a baseline which we denote as *Feature Maps*. Specifically, we encode an image $I$ using the CLIP vision encoder $\phi^v(I)$ to extract features $f \in \mathbb{R}^{N \times C}$. Note that the $N$ tokens can be reshaped into a 2-D feature activation map of shape $(H/P, W/P)$ where $H$ and $W$ are the height and width of the image $I$ and $P$ is the patch size of $\phi^v$. We take the corresponding tokens for each neuron in $C$ as the possible visual concepts. For ViT-B/16, $C = 768$ which amounts to 768 different visual concepts. We calculate the norm across tokens for

each neuron, and keep the top-k neurons with the highest norm. In order to label these neurons with a textual description, we build a simple concept labeler module to classify features into the $D$ textual descriptors. Note that we use the same facet of features and concept descriptors as in our multimodal explanations. During training of the concept labeler, we simulate the feature map labeling scenario where a part of the feature activation map is active. We implement this using DropBlock [17], an effective dropout technique [58] applicable to CNNs which drops a group of neighboring patches rather than individual patches. We use the per-class descriptors as ground-truth descriptors and train the classifier with Binary Cross-Entropy Loss on the full ImageNet training set, and validate it on the ImageNet validation set. An overview of this process for both training and concept labeling is shown in Figure 5. The feature activation map of the neuron with the highest norm usually identifies the main object in the scene, and acts similar to the Fiedler eigenvector we use in Section 3.1 of the main paper, while the remaining neurons act as the visual concepts we use. We show examples of this from 6 different images in Figure 6, with the left-most feature activation map representing the highest spatial norm. For each feature map, we normalize its values to be in the range between 0-1, and mask out values lower than a threshold of 0.9. We discover that neuron 689 (highest spatial norm) always encodes the main object, while other neurons are responsible for encoding visual concepts related to the object. The classifier attains a top-all accuracy of 65.76%, where "top-all" denotes the accuracy when all textual concepts of a class are accurately predicted. However, it is important to acknowledge that achieving high accuracy in this context is challenging due to the potential applicability of textual concepts from images of one class to images from another class. For instance, textual concepts such as *large eyes* may be relevant to numerous ImageNet classes, potentially exceeding 200 classes, yet they are only designated as one of the ground-truth concepts for 34 classes. Consequently, while predictions remain technically correct, they are penalized in terms of accuracy. Hence, we assess accuracy at the top-10 level rather than top-all, which involves considering the top-10 predicted textual concepts and measuring their alignment with the ground-truth concepts. This evaluation yields an accuracy of 83.84%.

## E   Evaluation of Concept-based Multimodal Explanations

In order to evaluate our multimodal explanations and compare them with the established baselines discussed in the previous section, we adopt evaluation metrics commonly used in the literature of explainable artificial intelligence. The deletion metric [42] starts with the original image and gradually removes image information (*e.g.,* pixels or patches) deemed most important by the explanation algorithm. The output score of the predicted class is then plotted as a function of the image information being removed, resulting in a curve. The Area Under the Curve (AUC) is then computed. A sharp decrease in this curve (low AUC) generally reflects a faithful explanation to the model. The intuition behind the deletion metric is that the removal of the "cause" will decrease the probability of the predicted class as important pixels are gradually removed. The insertion metric follows a similar process but in reverse, starting with a baseline image which removes the visual information of the image (*e.g.,* a heavily blurred version of the image), and gradually adds pixels or patches. In this case, a higher AUC indicates a more faithful explanation.

To adapt these metrics to our multimodal explanations and to the baseline methods, we perform the following steps: Initially, we rank the concepts in descending order of importance. Since each multimodal (visual-textual) concept is considered as one entity, the scores of the two modalities should be interdependent and mutually influential. We therefore compute the product of two similarity scores. The first score represents the CLIP visual similarity between the image and the visual concept, and the second represents the CLIP visual-language similarity between the image and textual descriptor. A visual concept and descriptor both representative will yield a higher score in this ranking process. Following the ordering of multimodal concepts, we proceed to add (in the case of insertion) or remove (in the case of deletion) the concepts. Given that the entities in our case are the concepts rather than individual pixels or patches, this process entails $L$ steps of insertion or deletion, where $L$ represents the number of concepts. Subsequently, we plot the zero-shot predicted class score against the concepts being added or removed. It is worth noting that removal of visual information typically involves zeroing out regions of the image, a practice that some recent studies [21, 46] have found leads to a change in the curve primarily due to the generation of Out-Of-Distribution (OOD) samples when portions of the image are zeroed out. To avoid this problem, we opt to blur the concept instead of zeroing it out. Figure 7 illustrates examples of both insertion and deletion steps along with their corresponding curves. We additionally incorporate the AccDrop metric proposed in [7] (denoted

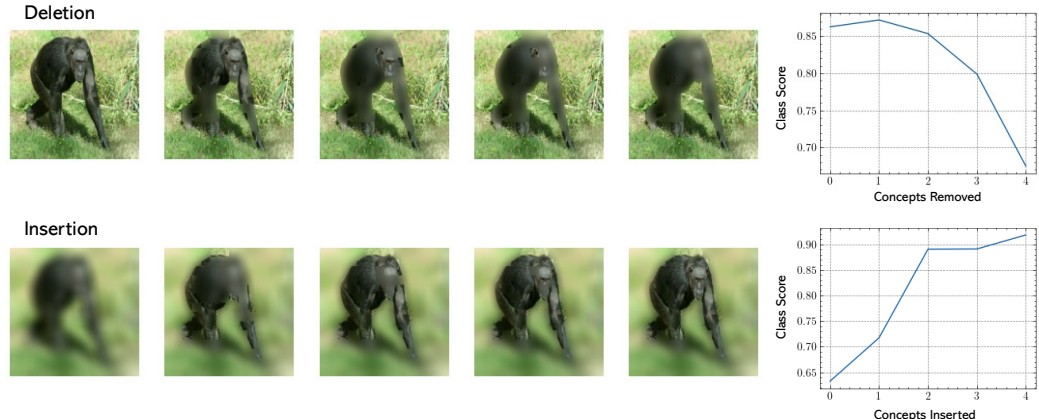

Figure 7: Deletion and Insertion steps generated by removing/adding the visual concepts (left) and inspecting how the zero-shot predicted class score changes (right). Last insertion step is removed to avoid clutter. Similarity score is multiplied by 2.5, following [20].

as the Positive Perturbation Test in [7]), which behaves similar to the deletion metric, but keeps track of the accuracy score rather than the predicted class score. By averaging results across the ImageNet validation set, this curve illustrates the decrease in validation accuracy as concepts are removed from the images. Lastly, we present the AccIncrease metric, which stands in contrast to AccDrop. Evaluation is conducted on the ImageNet validation set, and the results are averaged.

## F  Visual Prompt Engineering to Encode Regions

Several works in the literature of CLIP [54, 63] propose to perform visual prompt engineering in the image pixel space directly to encode a specific region of interest rather than encoding the whole image. These works mainly tackle the task of zero-shot referring expression comprehension and segmentation. In the simplest case, one could crop the region of interest from the image, and only feed that region to CLIP. This approach [59], however, removes contextual information and blurs the resulting cropped image, since the small crop has to be resized to the larger image size which CLIP expects. Recently, [54] show that by simply drawing a red circle on the region of interest, the CLIP vision encoder's attention shifts towards that region, effectively encoding it while also preserving contextual information and preserving spatial information. This work greatly outperforms the cropping-based approach. A concurrent work [63] further confirms this by drawing a more fine-grained boundary rather than a circle, but necessities the need for a strong fine-grained region selector such a SAM [26]. So we decided to use the circle-based approach.

In addition to the red circle annotation, we follow [54] and additionally include grayscaling and blurring outside the region of interest (referred to as Reverse-Grayscale and Reverse-Blur in [63], respectively), and then average the features from the vision encoder of all three visual prompts as the visual representation of the region. We follow the optimal hyperparameters from [54] and set the circle thickness to 1.

## G  MI on other classification datasets

In the main paper, we demonstrated experiments on the ImageNet dataset. In this section, we perform analysis on additional classification datasets. We first consider Places365 [65], which is a scene recognition dataset composed of 365 different scene locations. It is well-established that zero-shot CLIP does not perform well on this dataset, with the highest attainable accuracy of around 44%. We therefore test whether our mutual knowledge formulation aligns with accuracy in this scenario as well. In Table 6, we show two families of ViTs. The first is grouped according to the model size with the pretraining data fixed, and the second grouped among the pretraining data which varies. As shown, both groups align well with AUC, which further confirms our formulation. Finally, we perform an analysis on the Food-101 dataset, which classifies different types of food into 101 categories. Here

we fix the model size (ViT-B) and vary the pretraining data. As shown in Table 7, higher data quality align well with AUC, which further confirms our formulation on this dataset as well.

Table 6: MI and its dynamics (AUC) on the Places365 dataset

| Model | Data Size | Top-1 (%) | MI | AUC |
|---|---|---|---|---|
| *Model Size* | | | | |
| ViT-B/32 | 400M | 38.25 | 8.252 | 2.658 |
| ViT-B/16 | 400M | 38.82 | 8.655 | 2.872 |
| ViT-L/14 | 400M | 39.67 | 9.154 | 3.427 |
| ViT-L/14↑ | 400M | 40.36 | **9.073** | **3.450** |
| *Pretrain Data* | | | | |
| ViT-B/32 | 400M | 38.25 | 8.252 | 2.658 |
| ViT-B/32-dcp | 1B | 41.92 | 8.454 | 2.681 |
| ViT-B/16-dcp | 1B | 42.36 | 8.212 | 2.683 |
| ViT-B/16-dfn | 2B | 43.99 | **8.549** | **2.900** |

Table 7: MI and its dynamics (AUC) on the Food-101 dataset.

| Method | Data Size | Top-1 | MI | AUC |
|---|---|---|---|---|
| ViT-B/32-dcp | 1B | 85.81 | 8.358 | 2.948 |
| ViT-B/16-dcp | 1B | 90.30 | 8.185 | 3.307 |
| ViT-B/16-dfn | 2B | 91.24 | 8.364 | 3.763 |

# H  Additional Qualitative Examples

We show additional Qualitative Examples of our multimodal concept-based explanations in the visual encoder along with the concept importance map in Figures 8 and **??**. Each distinct visual concept is denoted by a different color, and the corresponding textual description is provided below, aligned with its corresponding color. Our multimodal explanations provide disentangled visually and textually fine-grained concepts of the visual features, such as the bird's nest, bill and body spots in the first two examples of Figure 8, and the towers, water structure and water jets of the fountain in the last example of Figure 8.

# I  Qualitative Examples of Vision-Language-Mutual Concepts

Additional Qualitative example of vision-language-mutual concepts are provided in Figure 10. In the first example, we see that the two mutual concepts are distinctive of the "flute", suggesting an effective encoding of the image-class inputs in the joint space. The second example presents a case where the mutual concepts are general, and not distinctive enough for the prediction of a "moped". This implies weaker shared knowledge for that prediction.

# J  Additional Related Work

We discuss an additional related work of analyzing concepts in contrastive vision-language models. The work of [64] investigates how well contrastive vision-language models such as CLIP learn textual concepts (denoted as primitive concepts) that together compose the predicted label (denoted as a compositional concept). This work first builds a Concept Bottleneck model for CLIP, and then analyzes it. Similar to [39], the bottleneck layer is built by the similarity of an image to all textual descriptors. A linear model is then trained on top of the similarity output to classify images. A prediction can then be directly explained by the linear combination of the textual concepts from the bottleneck layer. Next, a classifier is trained on the binary ground-truth textual concepts (denoting this model as Oracle-Prim), achieving almost 100% accuracy and validating the hypothesis that a linear compositional model can be learned from the ground-truth primitive concepts. The quantification of

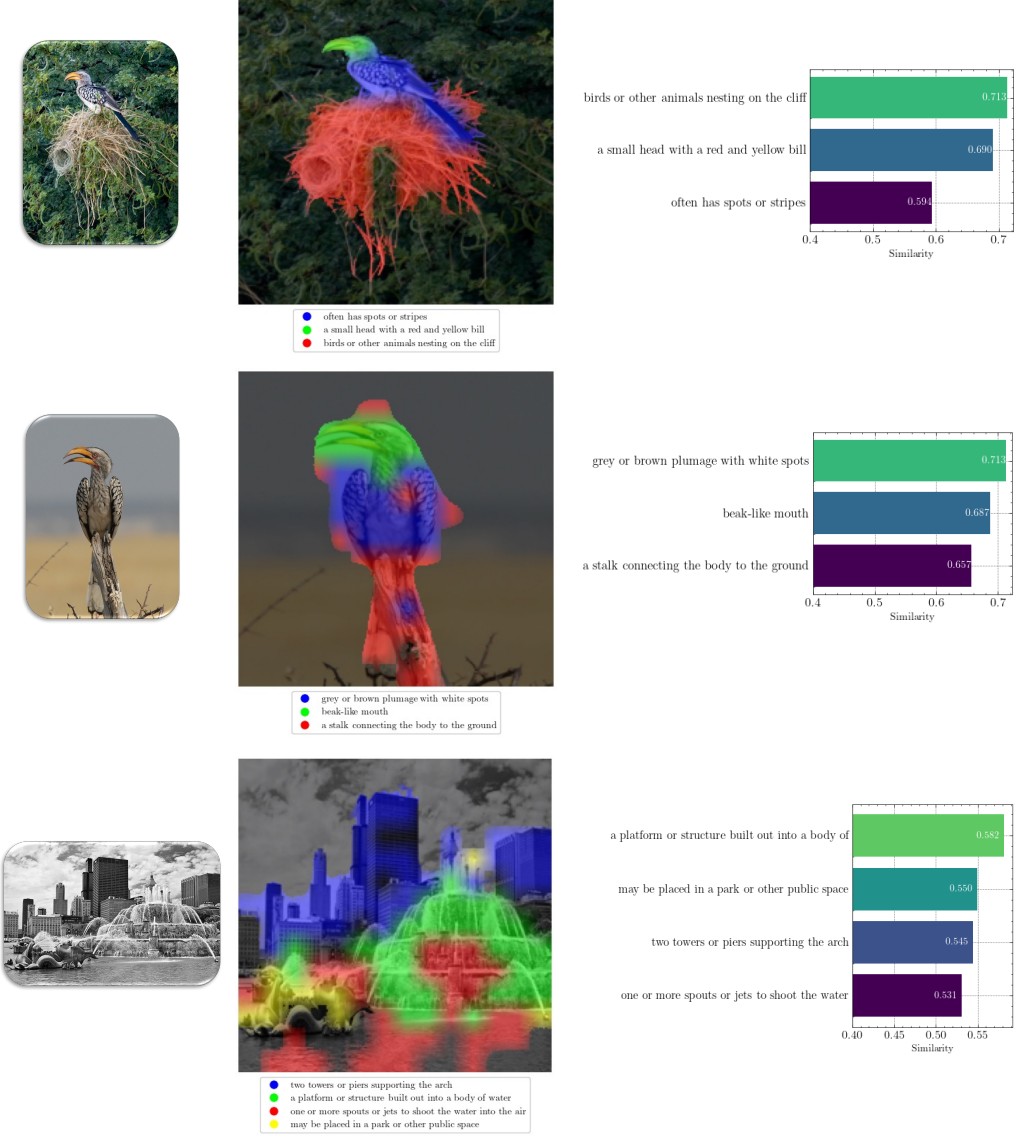

Figure 8: Additional Qualitative Examples of our multimodal concept-based explanations in the visual encoder, along with the concept importance map on the right

primitive concepts learned is measured by the difference between the classifier weights of Oracle-Prim (the learned classifier weight values when trained and fed with the binary ground-truth concepts) and the bottleneck model (the learned classifier weight values when trained on the similarity output of CLIP but fed with the binary ground-truth concepts). A smaller difference implies better performance of the bottleneck classifier. Note that this measure does not depend on the input, since in both cases the classifier is fed with the ground-truth textual concepts. They find that the difference is high. However, when the bottleneck classifier is fed with the similarity output from CLIP (what it was trained on) rather than the ground-truth concepts, the difference becomes smaller (better), implying that the classifier is utilizing irrelevant primitive concepts for prediction. To confirm this, they further evaluate the accuracy between the Oracle-Prim model and the ground-truth concepts, and show that it is almost perfect ($\approx$97%). Then they evaluate the accuracy between the bottleneck classifier and the ground-truth concepts, and show that it is very low, even approaching 0%. The final take-away message is that contrastive vision-language models do not learn primitive concepts well.

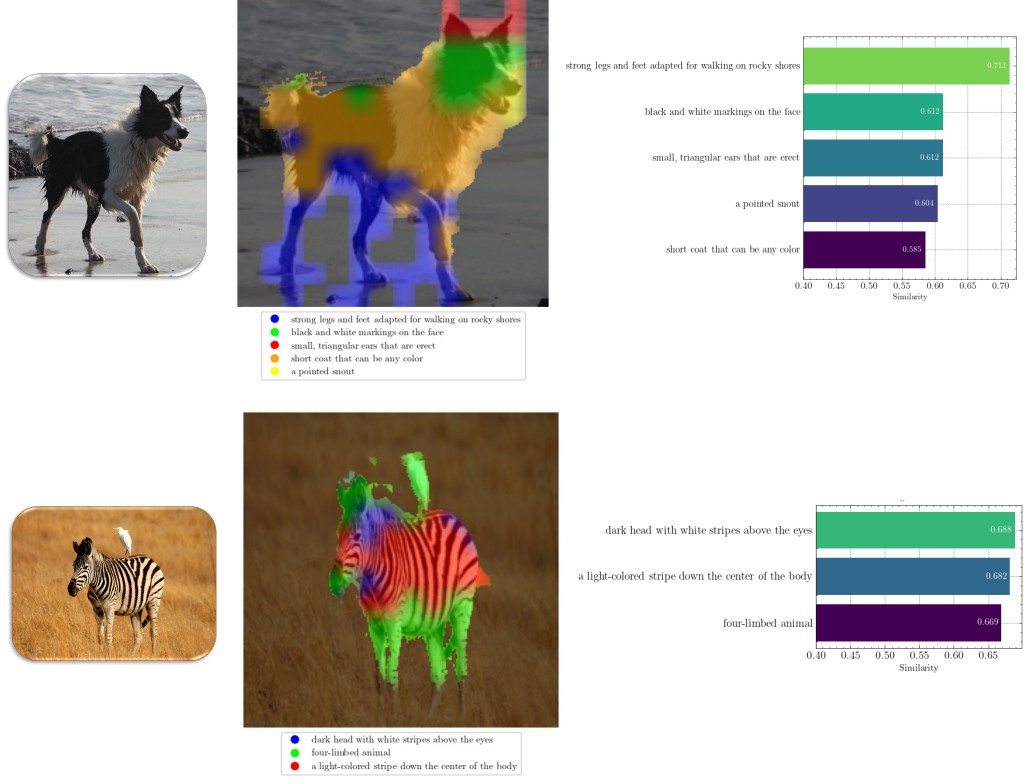

Figure 9: Additional Qualitative Examples of our multimodal concept-based explanations in the visual encoder, along with the concept importance map on the right

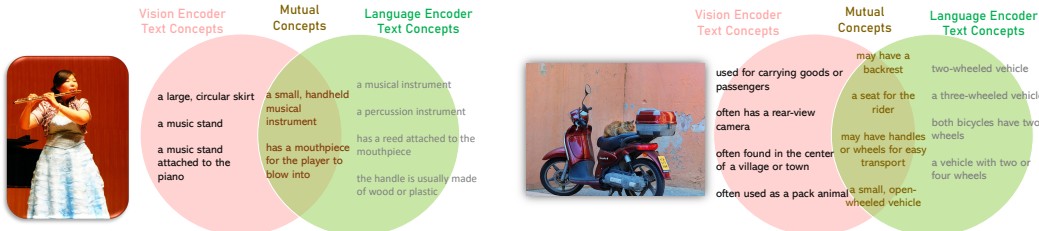

Figure 10: Qualitative Examples of our vision-language-mutual concepts

However, it is worth noting that this work only studies the joint feature space and using fine-grained datasets such as CUB [62] in which CLIP is shown to perform poorly. For example, the best CLIP model (ViT-L/14@336px) achieves a 49.5% zero-shot accuracy on the Birdsnap dataset [4], a fine-grained dataset for bird classification. Hence, this conclusion cannot be extrapolated to other datasets such as ImageNet, a general coarse-grained dataset encompassing a diverse set of 1000 objects from the real-world. Moreover, since this work is based on Concept Bottleneck Models, it inherits the same problems discussed in Section D.1. Finally, this study measures how well can CLIP dissect concepts from the joint feature space directly when explicitly trained to do so, while we study the degree of shared knowledge between the vision and language models which eventually leads to the alignment in the joint feature space.

## K  Additional Implementation Details

In section 3.1, we set $k = 500$ and $\tau = 1$. For analyzing the mutual information and its dynamics in Section 4.1 in the main paper, we set the number of concepts $L = 5$ and consider the top 3 textual

concepts for each visual concept. This results in a maximum of $5 \times 3 = 15$ textual concepts in the vision encoder, which are used for analyzing the MI dynamics. All baselines use also 5 concepts. For the CRF, we use the implementation from [29] and leave all parameters as their default settings. The original OpenAI CLIP models are provided from `https://github.com/openai/CLIP`.

For the application of CLIP zero-shot image classification with descriptors used in evaluating the effectiveness of our multimodal explanations, we use the prompt from [47]: `how can you identify a <class label>. Distinctive and physical features describing it is <descriptor>` for both the baseline methods [36, 43] and our method. This prompt has shown strong performance when class labels are paired with descriptors.

All experiments are ran on a single NVIDIA RTX3090 GPU. Experiments on the full ImageNet validation set require around 10-11 hours for base ViT models and 20 hours for large ViT models.

# L   Language Encoder Prompts for Classifiers and Descriptors

The prompts we use for the language encoder for constructing the zero-shot classifiers are shown in Table 8 and are averaged for each textual representation of the class [CLASS].

Table 8: Prompt templates for ImageNet, Places365, and Food101 datasets.

| Dataset | Prompt Templates |
|---|---|
| ImageNet | itap of a [CLASS].
a bad photo of the [CLASS].
a origami [CLASS].
a photo of the large [CLASS].
a [CLASS] in a video game.
art of the [CLASS].
a photo of the small [CLASS].
a photo of a [CLASS]. |
| Places365 | a photo taken in an [CLASS].
a photo of a [CLASS].
a scene taken in a [CLASS]. |
| Food101 | a photo of [CLASS], a type of food. |

The prompt we use for the language encoder for constructing the descriptors classifier is:

`a photo showing [DES]`, where [DES] represents the descriptor.

