# OpenReview forum: "Interpreting and Analysing CLIP's Zero-Shot Image Classification via Mutual Knowledge"
_NeurIPS.cc/2024/Conference — NeurIPS 2024 poster_

### Official Review · Reviewer_wbb9 · 2024-07-08

**Soundness:** 3
**Presentation:** 2
**Contribution:** 2
**Rating:** 6
**Confidence:** 4

**Summary:**

The paper investigates Contrastive Language-Image Pretraining (CLIP) for zero-shot image classification by exploring the mutual knowledge between visual and textual modalities. The study examines which concepts are commonly learned by both CLIP encoders and how they influence the shared embedding space. Using textual concept-based explanations, the researchers analyzed 13 different CLIP models, varying in architecture, size, and pretraining datasets. The approach provides insights into zero-shot predictions.

**Strengths:**

- Introducing mutual information is reasonable.

- The method is simple and effective.

- The authors provide comprehensive experiments and analysis.

**Weaknesses:**

- The introduction or Figure 1 does not clearly explain the motivation of the paper, making it difficult for me to understand.

- There is extensive literature on the generalizability (zero-shot) of CLIP, and this paper lacks comparison and discussion with the latest works.

**Questions:**

- What impact does the method proposed in this paper have on other fields (e.g., domain generalization)?

- Figure 3 and Figure 5 are not clear enough.

- Table 3 is not centered-aligned.

- Some tables in the appendix are missing top borders.

**Limitations:**

See Weaknesses and Questions.

---

> ### Author Rebuttal · Authors · 2024-08-06
>
> Dear Reviewer,
> Thank you for your time and effort in reviewing our paper, and for the positive feedback. We address your concerns below:
>
> > The introduction or Figure 1 does not clearly explain the motivation of the paper, making it difficult for me to understand.
>
> Our main objective is to interpret what the visual and language encoders of CLIP models learn in common, for the task of zero-shot image classification, through the use of textual concepts (short descriptions of natural language such as "a long snout", "feathered ears", "pointy ears"). Our approach uses textual concepts because: 1) they are human friendly and easily understood, and 2) each concept can be mapped to an integer using a dictionary of predefined concepts. For instance, the descriptor "a long snout" can always be mapped to the integer 0, "feathered ears" to the integer 1 and "pointy ears" to the integer 2. Once we have two sets of random variables representing those discrete integers (the textual concepts), one for the visual encoder and one for the textual encoder, then we can easily calculate the mutual information in the discrete space, which is fast, efficient and reliable (no need for approximations as in the continuous space).
>
> In order to achieve this, we need the two CLIP encoders to output random variables in the *same* space. In our case, we need both the visual and textual encoders to output *textual concepts*. To achieve this, we first convert the visual features into discrete visual units (visual concepts). In Figure 1b (top), these visual concepts are represented by the blue, red, green yellow and orange object parts. Once we have those, we annotate each visual concept with a textual descriptor (Figure 1b - bottom). Up until now we have the visual concepts represented as discrete textual concepts. Now we need to convert the language features for the zero-shot prediction to textual concepts. We do this by considering the zero-shot prediction as the center of a cluster in the joint space (green point in Figure 1c). This means that the points in that cluster (grey points) directly relate and explain the zero-shot prediction, and are considered as the textual concepts for that prediction. Now we have the textual concepts for the visual and language encoder, and we can see which concepts are common to both encoders. This can be seen in the Venn Diagram (Figure 1d). The left textual concepts are the ones specific to the visual encoder only, and the ones at the right are those that are specific to the language encoder. The middle ones (mutual concepts) are the *common* textual concepts that both encoders have, and we use them to calculate the mutual information and its dynamics. These are also what influences the visual-language points (green and orange points in Figure 1a) to be closer or further away in the joint space. This also allows us to see how the two encoders influenced each other when making the zero-shot prediction.
>
> We will make sure to refine the introduction and caption of Figure 1 so that it better describes the motivation as written above.
>
> > There is extensive literature on the generalizability (zero-shot) of CLIP, and this paper lacks comparison and discussion with the latest works
>
> We assume that the reviewer is referring to the line of work on few-shot CLIP adaptation. Kindly note that these works are not relevant to our study, as they are based on parameter-efficient tuning/prompt tuning with few-shot examples to make CLIP better adapt to other domains. They do not involve any descriptors, and are based on learning from a few set of ImageNet examples. On the other hand, the purpose of our experiments in Table 2 is to show that the multimodal concepts and descriptors identified, are a reliable source of input for mutual information analysis. Therefore, the two works in the literature that involve using descriptors as additional input prompts to the language encoder, are [35,42] (cited in our manuscript) and we have compared with them and greatly outperformed them. Therefore, we believe that there is no overlap with the line of work on CLIP adaptation.
>
> > What impact does the method proposed in this paper have on other fields (e.g., domain generalization)?
>
> Thank you for raising this question. Our work offers a human-friendly approach to interpreting and understanding zero-shot predictions and CLIP models, and explaining why their performance increases or decreases (e.g., what mutual concepts caused this). Furthermore, the quantitative analysis of mutual information allows us to discover new relationships which can aid in fields such as model and architecture design, data curation and filtering as well as prompt and model selection (these are detailed in lines 277-286 of the manuscript). As for domain generalization, we could potentially use our approach to see if mutual concepts identified for a CLIP model using one dataset (e.g., ImageNet), can aid the performance of the model on other datasets without any training. While this is an interesting experiment, we leave it to future work, as it deviates away from the central hypothesis of our study. Finally, apart from the mutual concepts and mutual analysis, our multimodal concepts in the vision encoder alone also offer significant gains in zero-shot classifications (Table 2), which are (most importantly) interpretable (e.g., we know which concepts caused improvements and how to manipulate the joint mutual embedding space).
>
> > Figure 3 and Figure 5 are not clear enough.
>
> Thank you this comment. We will revise the figures and enhance the clarity by making the curves thicker, and the text in the figures bigger.
>
> > Table 3 is not centered-aligned, and some tables in the appendix are missing top borders.
>
> Thank you for this comment. We will make sure to fix this issue in the revised manuscript.

---

> > ### Comment · Reviewer_wbb9 · 2024-08-12
> > **Official Comment by Reviewer wbb9**
> >
> > Thanks for the authors' response. I kept the original score.

---

### Official Review · Reviewer_Whov · 2024-07-10

**Soundness:** 2
**Presentation:** 1
**Contribution:** 2
**Rating:** 5
**Confidence:** 3

**Summary:**

The work deals with explainable artificial intelligence (XAI) in a multimodal (text-image) context. The proposed approach consists of first identifying the most important patches of a collection of images, and associating them to a textual description through CLIP. Hence, relying on a textual description of both the images and some textual description of the classes, the approach computes their mutual information for a varying number of visual concepts (still described by text), resulting in a curve from which it derives a score (AuC) that represent the "mutual information dynamics". The approach is evaluated on Imagenet for the task of zero-shot classification.

**Strengths:**

* Analyzing explicability through the prism of mutual information between visual and textual features is original and potentially fruitful. The proposed approach consists to project the visual part to textual one but one could imagine further works

* The paper reports a quantitative estimation of the explainibility, using four metrics previously defined in [7, 41]. The proposed approach exhibits good performance in comparison to three recent methods, some having been adapted to the multimodal case.

* The method is tested with several visual encoders, from several types (visual transformer and several convolutional neural networks).

**Weaknesses:**

* the presentation of the method in Section 3 is quite hard to follow:
  - first, it starts by "CLIP [65] formulates image classification as a retrieval task by using the textual class labels (...) and encoding them with the language encoder." which is a very imprecise way of putting things. CLIP does not "formulate" any task, it is a neural model that learns visual representation with natural language supervision. It can be instructed in natural language to perform image classification but it requires a specific approach (textual prompt) in each case.
  - the notations on lines 107-114 may be clearer. For example, it starts by considering an image $I$, then describes CLIP by splitting it between an encoder and a final linear layer that are no longer used in the following, to finally considering the similarity $s(i,j)$ between an image (that was $I$ at the beginning) and a text (never introduced)
  - lines 115-121, the text is unclear. The sentence "a unified set of $\mathcal{D}$ class-agnostic textual descriptors of length D" seems to mean that there are $\mathcal{D}$ 'descriptors' and that each is of length D. However, the following leads to doubt about that since the manuscript reports "D = 4,229 after discarding repetitive descriptors across the entire pool" suggesting that D is the number of 'descriptors'. Moreover, on line 118 "Concepts in $\mathcal{D}$" suggests that $\mathcal{D}$ is now the name of the set and not the number of concepts. After several readings, I think that the authors refer to the "length of the set" as Python programmer while one should consider the *size* of the set, meaning that D=4,229 is the number of elements in the set $\mathcal{D}$.
  - it is nevertheless still unclear what is a 'descriptor'. On line 120, an example of 'textual descriptor' is given ("can be hung from a tree") suggesting that it is actually a *description*. However, it is unclear how these descriptions are obtained, the only hint being on line 115: "We utilize an LLM to generate descriptors...". How these LLMs are used? From which data (what is the prompt)?  While lines 102-114 introduce many notations just before, the names of the classes (one guesses $\mathcal{Y}$ on line 103) are no longer used in the following
  - on line 134, it is not clear what is "the zero-shot predicted class" nor how it is obtained. After some consideration on mutual information (lines 122-126) the overview of the method starts with a "given set of images" (line 127) thus it is unclear to which image (images? patches?) this "zero-shot predicted class" relates to.
  - In the same vein, the (same?) "zero-shot prediction of CLIP" at the beginning of section 3.2 should relate to an image but it is not clear which one
  - Line 205, how the "importance to the image" is determined?
  - Last but not least, it may be worth explaining how the "explanation" is provided by the proposed model once the "mutual information dynamics" is obtained

* Similarly, Section 4 starts by explaining that an evaluation was conducted but the reader does not know for which task nor on which dataset; ImageNet is used for training (line 222) thus one guesses that it deals with zero-shot image classification (cf. title of the article) on the validation set of IM-1000? However, no implementation is provided for the experiment which results are reported in Table 1: which visual encoder is used (several are considered in the following)? how are they trained? Are the Vit/CNN pre-traind on some data? Which textual encoder is used? what is the LLM used to get the textual description of the classes and how is it done in practice? Moreover:
- the manuscript refers to [7] to explain the metrics "Accuracy Drop" and "Accuracy Increase" (lines 224-225). However, in [7] the word "Increase" does not appear and the only occurrence of the word "Drop" does not refer to a metric. It is only in Appendix F that the reader can understand that the name of the metrics was changed. Moreover, if the "adaptation" of the metric is explained (still in the appendix) for the proposed method, it is not clear whether such an adaptation is required for the other methods (and whether it is the same as for the proposed approach). Previous works such as [37, 38] report the accuracy of each model, which seems relevant.

* Previous works such as [18, 37, 38] usually evaluate their approach on several datasets. However, one must admit that they do not report quantitative evaluation in terms of explainibility.

* On the opposite, it would have been relevant to compare the approach to those that specifically deal with XAI such as [a,b,c], in the vein of [d] which is a reference that could be cited. More generally the proposed work dealing with prototypes, it could be relevant to cite some of the works that adopted such scheme since [e].

[a] Wang, C., Liu, Y., Chen, Y., Liu, F., Tian, Y., McCarthy, D. J., Frazer, H., and Carneiro, G. (2023). Learning support and trivial prototypes for interpretable image classification. ICCV

[b] Lei, Y., Li, Z., Li, Y., Zhang, J., and Shan, H. (2023). Lico: Explainable models with language image consistency. NeurIPS
Information Processing Systems

[c] Wan et al (2024) Interpretable Object Recognition by Semantic Prototype Analysis, WACV

[d] Chen et al (2019) This Looks Like That: Deep Learning for Interpretable Image Recognition. NeurIPS

[e] Saumya Jetley, Bernardino Romera-Paredes, Sadeep Jayasumana, and Philip Torr (2015) Prototypical Priors: From Improving Classification to Zero-Shot Learning. BMVC

**minor points**
- the text in Figure 1 is barely legible. On a standard A4 page, it is hard to read the list of concepts above (d).
- reference [21,52] are incomplete
- reference [29] is wrong, it has been published in NIPS 2012. Actually, a 2017 paper was published in Communication of the ACM, but it is clearly written (bottom right) that the original paper was published at NIPS 2012.
- the date is missing in [30]
- [49] both refer to the article and the (arxiv) preprint
- line 224: low --> lower
- The derivation in Appendix C is not very useful as the result is well-known for uniform distributions. However, a discussion on such uniformity may nevertheless be relevant.

**Questions:**

Globally, many points are unclear as explained in the "weakness" section. Several implementation details should be reported in the manuscript, in particular which textual encoder is used? what is the LLM used to get the textual description of the classes and how is it done in practice? As well, how the "explanation" is provided by the proposed model once the "mutual information dynamics" is obtained?

**Limitations:**

Two limitations are discussed in the Appendix (section D), one relating to the visual concepts that are not located in an image and the other to the type of tasks addressed (classification only). It nevertheless ignores the limitations linked to the usage of an external LLM to create the description of the classes, the limited evaluation (on one dataset only) not to mention the influence of bias in data in practical cases (that is an important issue to consider for the works dealing with explanations)

---

> ### Author Rebuttal · Authors · 2024-08-06
>
> Dear Reviewer,
> Thank you for your time. We will address all your concerns here and will include all of them in the revised manuscript.
>
> > it is not clear what is "the zero-shot predicted class" nor how it is obtained
>
> The process of how CLIP performs zero-shot prediction is already described in lines 102-106. We will clarify it here better: Each class name is converted into a textual prompt of: "an image of a {class name}", and encoded with the language encoder of CLIP. The image is then encoded with the vision encoder of CLIP. Since CLIP maps both images and text into a joint embedding space, the text prompt with the highest similarity to the image is considered as the predicted class.
>
> > "CLIP formulates image classification as a retrieval task" is a very imprecise way of putting things
>
> We used the word "formulate" because CLIP is a vision-language encoder rather than an "instructable decoder" such as GPT. In CLIP, the task of image classification is formulated as (converted to) an image-text retrieval task in which we retrieve the closest text to an image. This description is standard in CLIP literature.
>
> > Which textual encoder is used?
>
> In CLIP literature, it suffices to describe the visual encoder of CLIP (e.g., ViT-B/16). Each visual encoder has its own associated textual encoder, which is defined by OpenAI or OpenCLIP. For OpenAI models, these can be seen in Tables 19-20 (last page) of the original CLIP paper: https://arxiv.org/pdf/2103.00020
>
> > limited evaluation (on one dataset only)
>
> Kindly note that we did perform evaluation on two other datasets than ImageNet, which are Places365 and Food101. These are shown in Tables 5 and 6 in the Supplementary material (page 23) and are also referenced in the main manuscript in lines 273-276. We have deferred them to the Supp. material due to space limitations.
>
> >The notations on lines 107-114 may be clearer.
>
> We have mentioned the linear projection layer to make a difference between the visual features used for extracting visual concepts (which we denote as $f$ and use later in Section 3.1) and the projected features used for similarity calculation (the output of $\psi^v$) as described in line 113 and used later in line 182 and Section 3.2). Furthermore, the text input $j$ is introduced in line 112. We acknowledge that this may not in the most clear way and can lead to confusion, and will introduce $j$ earlier in line 107. We will also change the similarity definition to $s(I,j)$
>
> > it is unclear what is a 'descriptor'
>
> We use the word "descriptor" to express a short textual description of a class (line 43). We also referred to it as a "textual concept". The word descriptor is usually preferred in the literature of CLIP to avoid confusion with long descriptions such as captions
>
> > Details of Baselines in Section 4
>
> These are mentioned in Section E of the Supplementary material. Please note that these are not the central hypothesis of our work, and are meant for showing that our multimodal explanations are a reliable source of mutual information analysis. That is why we deferred them to the Supp. material. MM-CBM and MM-ProtoSim are trained with Image Classification objective and their performance is shown in Table 4. The CLIP model we used here is CLIP ViT-B/16 (mentioned in the caption of Table 4). Please also note that all baselines, and our method use the same set of descriptors (as mentioned in line 619)
>
> > line 127, it is unclear to which image this "zero-shot predicted class" relates to
>
> Each image has its own zero-shot predicted class
>
> >lines 115-121, the text is unclear
>
> You are right in the sense that the sentence "a unified set of $\mathcal{D}$ class-agnostic textual descriptors of length D" may be confusing. We will therefore clarify this better as: "$mathcal{D}$ is the set containing all task agnostic descriptors and its cardinality (the number of descriptors it contains) is D, that is, $D=|\mathcal{D}|$
>
> > How these LLMs are used? From which data (what is the prompt)?
>
> In the supplementary material (line 843), we mentioned that we directly used the descriptors provided by [35] which uses GPT 3.5 with the prompt: "What are useful visual features for distinguishing a {category name} in a photo?". We have also provided ablation experiments on the prompt and LLM (please refer to the Table in the response for Reviewer BMwb).
>
> > how the explanation is provided by the proposed model once the mutual information dynamics is obtained
>
> The provided explanation is the Venn Diagram shown in Figures 1 and  6. The mutual information dynamics process takes as input this explanation, and delivers a quantitative analysis. Our work therefore provides both a qualitative visualization of the mutual information, and a quantitative analysis of it
>
> > Is the metric adaptation the same as for the proposed approach? Previous works such as [37, 38] report the accuracy which seems relevant
>
> All baseline methods and our proposed method yield multimodal explanations and are all evaluated on the same set of metrics, with exactly the same way and using the same visual features. We also did report the accuracy of our multimodal baselines and compared with both [37, 38], and we showed that our multimodal baselines even outperforms these works (please refer to Table 4, page 19 in our manuscript)
>
> > Other approaches such as [a,b,c], and references that could be cited
>
> Kindly note that this line of work is very different from ours as 1) they only provide a single-modality explanation, 2) they are explicitly trained to generate prototypes, and 3) they do not scale to ImageNet; they are typically applied on CUB birds dataset. These three points were the criteria to select our baseline methods. Particularly, we selected baselines that scale to ImageNet and are not explicitly trained to generate prototypes (LF-CBM, ProtoSim, Feature Maps) and adapted them to the multimodal case. We will, however, cite the works you mentioned and include this discussion

---

### Official Review · Reviewer_BMwb · 2024-07-19

**Soundness:** 3
**Presentation:** 3
**Contribution:** 2
**Rating:** 6
**Confidence:** 4

**Summary:**

The authors propose to interpret CLIP models for image classification from the lens of mutual knowledge between the image and text encoders of CLIP. Specifically, the authors use textual concepts as the common medium of the two modalities by mapping visual concepts to textual concepts. The authors then calculate mutual information in the shared textual space for analysis. Extensive experiments on a set of CLIP models demonstrate the effectiveness of the proposed method in understanding CLIP’s zero-shot classification decisions.

**Strengths:**

-	The proposed method is well motivated. The idea of mapping visual concepts to textual concepts and using the mutual concepts in the shared textual space for interpretation is interesting.
-	The paper is generally well-written and easy to follow.
-	The authors provide a comprehensive analysis on a broad set of CLIP models varying in architecture, size, and pre-training data. The experiments are extensive and the results are promising.

**Weaknesses:**

-	In L115, the authors use LLM to generate descriptors for all classes. However, it seems that some details are missing here. How exactly the descriptors are generated? What specific prompts and LLM are used? Is there any filtering mechanism to remove noisy text descriptions? How to maintain the relevance and diversity of the generated descriptors for each class? It seems that the design of prompts tends to affect the generated concepts a lot. The authors should also provide ablations on the effect of generated concepts as well.
-	Given the noisy nature of PCA/K-means used for visual concepts, how do the authors filter these failure cases?
-	In Table 1, the proposed method does not perform well on the Insertion Metric. Could the authors provide the justification for this?

**Questions:**

See the questions mentioned above. I am leaning towards borderline accept and hope the authors could address my concerns during the rebuttal.

**Limitations:**

The authors have addressed the limitations in Sec. D (supplementary material), which looks good to me. It would be better to discuss some broader societal impacts of the work as well.

---

> ### Author Rebuttal · Authors · 2024-08-04
>
> Dear Reviewer, We thank you for your time and effort and for the valuable feedback you provided. We will address each of your concerns below:
>
> > How are the descriptors generated? What specific prompts and LLM are used?
>
> In the supplementary manuscript (line 843), we mentioned that we directly used the descriptors provided by [35] (reference in our manuscript). This work uses GPT-3.5 for generating the descriptors, using the following prompt: "What are useful visual features for distinguishing a {category name} in a photo?". This work also uses an in-context example to instruct the LLM to generate structured descriptors (short, distinctive). We will include these details in the manuscript in Section 3.
>
> > How to maintain the relevance and diversity of the generated descriptors for each class? The authors should also provide ablations on the effect of generated concepts as well.
>
> We find that the generated descriptors are of good quality. To show this, we have conducted an ablation study on different prompts, as well as different LLMs. For each (LLM, prompt) experiment, we measured 1) the zero-shot top-1 and top-5 accuracy: these measure the relevancy of the descriptors to CLIP, and a higher accuracy implies more relevant descriptors to the class. 2) Intra-Class Diversity: this is the cosine similarity between the different descriptors of a given class, averaged over all ImageNet classes. We used the Sentence Transformer language encoder to encode the descriptors. Note that, the lower the similarity is, the more diverse the descriptors are. Therefore, lower is better. Finally, 3) Inter-Class Diversity, measures the diversity of descriptors across different classes rather than across a single class. We considered 4 LLMs: GPT-3.5, GPT-4o-mini, GPT-4o, and the latest Llama3.1-8B-Instruct. We also considered an ensemble of 2 LLMs: GPT-3.5 and GPT-4o-mini, where GPT-3.5 provides context to GPT-4o-mini, and GPT-4o-mini answers according to its own knowledge as well as the context.
>
> Moreover, we considered 4 prompts (P):
>
> P1: "What are useful visual features for distinguishing a {category name} in a photo?"
>
> P2: "What are the distinctive and physical features of a {category name} ?"
>
> P3: "What specific attributes distinguish a {category name}?"
>
> P4: "Which physical features and attributes make a {category name} different from others of the same type?"
>
> Here are the results using a ViT-B/16:
>
> |Prompt|LLM|Top-1|Top-5|Inter-Class Diversity|Intra-Class Diversity|
> |-|-|-|-|-|-|
> |P1|GPT-3.5|67.93 |91.45|0.345|0.206|
> |P1|GPT-4o-mini|68.39|91.74|0.236|0.172|
> |P1|GPT-4o|68.42|91.66|0.246|0.175|
> |P1|Llama3.1-8B-Instruct|68.19|91.56|0.263|0.184|
> |P2|GPT-4o-mini|68.35|91.69|0.236|0.164|
> |P3|GPT-4o-mini|68.39|91.78|0.231|0.152|
> |P4|GPT-4o-mini|**68.56**|**91.83**|**0.228**|**0.151**|
> |P4|GPT-3.5 + GPT-4o-mini|68.40|91.68|0.236|0.159|
>
> We found that P4 with GPT-4o-mini provides the best results in terms of all metrics. However, the effect is very marginal (e.g., 0.63 accuracy improvement, and 0.11 diversity improvements). Therefore, the experiment we used in our work (P1, GPT-3.5) is reliable. We will add these experiments in the manuscript. Finally, we would also like to bring to your attention that our method produces diverse textual descriptors, as evidenced by the ablation studies of Optimal Transport (please refer to Table 2, page 14 in our manuscript).
>
> > Is there any filtering mechanism to remove noisy text descriptions?
>
> While this is not done on the LLM side, it is performed during our methodology when assigning the textual descriptors to visual concepts (please refer to footnote 1 in Page 5 of our manuscript). We will clarify this aspect better in the manuscript.
>
> > In Table 1, the proposed method does not perform well on the Insertion Metric. Could the authors provide the justification for this?
>
> We thank you for raising this point. Upon inspecting several samples, we found that the baseline "MM-CBM" is generally characterized by a sparse set of patches that are deemed as important. This is due to the attention mechanism which gives most of the weight mass to a small set of patches, as typical in most attention mechanisms. Below we report the average number of patches (denoted as ImpPatch) deemed as important (rounded to the nearest integer), across a set of ImageNet validation classes, for both our method and the baseline MM-CBM.
>
> |Method|ImpPatch|
> |-|-|
> |MM-CBM |6.0|
> |Ours |81.0|
>
> This essentially means that each concept in MM-CBM is characterized by 1 or 2 patches. Therefore, during the insertion process, the generated curve increases very sharply. On the other hand, for our method, the concepts are a decomposition of features (all principal components reconstruct back the feature space). This implies that each of those components (the concepts) plays an important role, and the mass is distributed across a much larger number of patches. This causes a gradual increase in the curve during the insertion process (rather than a sharp increase in the case of MM-CBM). Sharp increases cause a higher AUC, which is why MM-CBM performs better on this metric. We will include this clarification in the manuscript.
>
> > Given the noisy nature of PCA/K-means used for visual concepts, how do the authors filter these failure cases?
>
> We have mentioned how we tackled this problem in Section D, page 16. We first experimented with alternative clustering algorithms such as DBSCAN and Hierarchical Clustering; nevertheless, this approach did not show improvements. Instead, a good approach has been to consider the largest connected region of that visual concept (as written in line 576), effectively ignoring the noisy scattered concepts. This alleviates the issue mentioned and does not have a negative effect on the association of the visual concepts to a textual descriptor (which is an integral part of the Mutual Information calculation). The only drawback in this case, is that the concepts are less visually appealing.

---

> > ### Comment · Reviewer_BMwb · 2024-08-12
> > **Official Comment by Reviewer BMwb**
> >
> > Thanks for the authors' response. The rebuttal well addressed my concerns. I would like to increase my score to 6. I suggest the authors incorporate all the discussions into the final version.

---

### Author Response · Authors · 2024-08-11

Dear Reviewers,
Thank you again for your time and effort in reviewing our paper. As the discussion period is coming to an end, we would kindly like to ask you to go through our rebuttal. We believe that we have sufficiently addressed all your concerns and sincerely hope for reviewers BMwb and Whov to raise their scores. Thank you and good day!

---

### Decision · Program_Chairs · 2024-09-25

**Decision:**

Accept (poster)

**Comment:**

This paper proposes a way to measure the relationship and mutual knowledge between the image and text encoders of CLIP. The paper received 2 weak accepts and 1 borderline accept final recommendations from reviewers. Positive points included the well-motivated and original approach, comprehensive analysis, and promising results. Negative points included some unclear motivation/details, weak performance on certain metrics, and missing comparisons. Most of the concerns were adequately addressed by the rebuttal. Overall, after carefully considering the paper, rebuttal, and discussions, the ACs feel that the paper's strengths outweigh its negatives, and therefore recommend accept. It is recommended that the authors incorporate the rebuttal points into the final version.